# Loss of PLA2G4E compromises synaptic structure and cognitive outcomes in mice

Sara Badesso[1], Marta Perez-Gonzalez[1], Sara Exposito[2], Maria Espelosin[1], Clara Cambria[3], Laura D'Andrea[4], Gabriele Imperato[4], Pedram Moeini[5], Africa Vales[5], Gloria Gonzalez-Aseguinolaza[5], Nico Mitro[4,6], Flavia Antonucci[3], Eduardo D Martín[2], Elena Marcello[4], Mar Cuadrado-Tejedor[1,7], Ana Garcia-Osta[1]

Given its potential role in supporting cognitive resilience, PLA2G4E has emerged as a compelling therapeutic target in the context of Alzheimer's disease (AD). However, its physiological functions in the central nervous system remain largely unexplored. In this study, we demonstrate that Pla2g4e expression peaks during early postnatal brain development, coinciding with the rapid formation of synapses. Loss-of-function experiments in primary neuronal cultures revealed that Pla2g4e expression is essential for proper dendritic development and neuronal maturation. In constitutive Pla2g4e knockout mice, we observed significant disruptions in the developmental profiles of cortical synaptic plasticity markers, accompanied by impairments in memory-related behaviors. Notably, the adeno-associated virus–mediated overexpression of PLA2G4E rescued memory deficits, highlighting its functional relevance in cognitive processes. Furthermore, selective deletion of Pla2g4e in excitatory neurons of the adult brain resulted in moderate memory impairments in aged animals, suggesting an ongoing role in synaptic maintenance. Together, these findings establish PLA2G4E as a key regulator of dendritic architecture, synaptic function, and cognitive performance, and highlight its potential as a gene therapy target for neurodegenerative diseases characterized by synaptic dysfunction.

## Introduction

Cognitive resilience, defined as the brain's ability to maintain function despite pathological challenges, has emerged as a critical focus in neuroscience research. Rather than concentrating solely on identifying risk factors for cognitive decline, growing attention is being directed toward understanding the mechanisms that promote resilience against dementia. Elucidating these protective factors offers promising opportunities for the development of novel therapeutic interventions (1, 2). In line with this perspective, previous studies from our laboratory identified Pla2g4e (encoding phospholipase A2 group IVE, also known as cPLA2ε) as a gene associated with cognitive resilience in experimental models of Alzheimer's disease (AD), suggesting a potential role of Pla2g4e in supporting synaptic integrity and cognitive function under neurodegenerative conditions (3).

Notably, this study demonstrated that adeno-associated virus (AAV)–mediated delivery of PLA2G4E into the hippocampus led to reverse the memory impairment observed in the APP/PS1 AD mouse model and counteract the loss of synaptic connections, supporting its potential as a therapeutic target (3). Similarly, in Celsr2 knockout mice, a model characterized by impaired spinogenesis and motor learning deficits, the overexpression of PLA2G4E resulted in increased spine formation rate and improvements in motor learning (4). In addition, PLA2G4E expression is up-regulated in primary neuronal cultures after bicuculline treatment, as well as after and upon chemogenetic activation of motor neurons, suggesting that this enzyme may modulate adult synaptic plasticity in an activity-dependent manner (3, 4). These findings highlight the potential of PLA2G4E in the establishment and/or stability of synaptic connections. Moreover, this phospholipase plays a major role in the tubular formation supporting the clathrin-independent endocytic pathway (5). This route is essential for rapid endocytosis triggered by specific stimuli, like in the case of synaptic vesicle recycling, which is crucial in synaptic plasticity (6, 7).

PLA2G4E exhibits modest phospholipase A2 activity but demonstrates robust $Ca^{2+}$-dependent N-acyltransferase activity (8, 9). Specifically, PLA2G4E catalyzes the transfer of a fatty acyl chain from the sn-1 position of phosphatidylcholine (PC) to the amino group of phosphatidylethanolamine (PE). This activity results in the formation of N-acyl-PEs (NAPEs) that serve as precursors to N-acyl-ethanolamines (NAEs). Recent research has highlighted PLA2G4E as the primary enzyme responsible for biosynthesis of NAPEs and subsequent NAEs in the mammalian brain. Both NAPEs and NAEs play crucial roles in a variety of cellular and physiological functions.

[1]Gene Therapy for CNS Disorders Program, Center for Applied Medical Research (CIMA), Instituto de Investigación Sanitaria de Navarra (IdiSNA), University of Navarra, Pamplona, Spain  [2]Instituto Cajal, Consejo Superior de Investigaciones Científicas (CSIC), Madrid, Spain  [3]Department of Medical Biotechnology and Translational Medicine (BIOMETRA), University of Milan, Milan, Italy  [4]Department of Pharmacological and Biomolecular Sciences "Rodolfo Paoletti", University of Milan, Milan, Italy  [5]DNA and RNA Medicine Division, Center for Applied Medical Research (CIMA), University of Navarra, Pamplona, Spain  [6]Department of Experimental Oncology, IEO, European Institute of Oncology IRCCS, Milan, Italy  [7]Department of Pathology, Anatomy and Physiology, School of Medicine, University of Navarra, Pamplona, Spain

Correspondence: mcuadrado@unav.es; agosta@unav.es

Their concentrations increase notably in response to ischemic events, contributing significantly to neuroprotection mechanisms (10, 11, 12, 13, 14). Interestingly, N-palmitoylethanolamide (PEA) and N-oleoylethanolamide (OEA) were found to mediate anti-inflammatory processes and enhance memory by activating several pathways, including those downstream of PPARα (15, 16). Interestingly, treatments directed to increase the levels of NAEs, such as the administration of the fatty acid amide hydrolase (FAAH) inhibitor URB597 (17, 18, 19), or the direct administration of PEA or OEA, improved memory in various animal models exhibiting cognitive deficits (14, 20, 21).

Furthermore, genetic and epigenetic variants within the *PLA2G4E* gene have been associated with potential risks for neurodevelopmental disorders, such as panic disorder, and atypical neurobehavior in preterm infants (22, 23). These findings suggest that variations, whether genetic or epigenetic, in this gene, might impact future behavior and cognition. Given the critical role of appropriate synaptogenesis and synapse homeostasis in normal brain function, our hypothesis is that PLA2G4E plays a crucial role during brain development.

In this study, we aimed to elucidate the physiological role of PLA2G4E in the brain by investigating its spatial and temporal expression patterns and functional relevance during postnatal development. We found that Pla2g4e expression peaks during early postnatal stages, coinciding with periods of intense neuronal differentiation and synaptogenesis. Functional experiments using primary neuronal cultures demonstrated that early disruption of *Pla2g4e* impairs dendritic arborization, synapse formation, and synaptic transmission. To explore the in vivo consequences, we used *Pla2g4e* knockout mice (*Pla2g4e$^{-/-}$*), as well as AAV-based gene manipulation strategies to modulate its expression. Behavioral assessments of *Pla2g4e$^{-/-}$* mice revealed deficits in memory and increased stereotyped behavior, phenotypes that mirror those observed in animal models of autism. Collectively, our findings highlight PLA2G4E as a critical regulator of brain maturation, with its dysfunction potentially contributing to neurodevelopmental and cognitive disorders.

## Results

### PLA2G4E expression increased during developmental stages

Recent studies highlighted that *Pla2g4e* mRNA expression in mouse brain is significantly higher in neonatal animals (at postnatal day 0, PND0) with respect to adults (11, 24), and that the $Ca^{2+}$-dependent N-acyltransferase activity follows a bell-shaped pattern that peaks in the first postnatal week (PND7) and varies depending on the brain region, showing the highest activity in the cortical regions (11).

To investigate the role of Pla2g4e in the central nervous system (CNS), and more specifically in synaptic plasticity, we characterized its protein expression levels in mouse cerebral cortex during the early postnatal period. Protein levels of Pla2g4e were assessed in 2% SDS protein extracts prepared from the prefrontal cortex of C57BL/6J mice euthanized at different postnatal days (PND1–60). As shown in Fig 1A, we observed a bell-shaped expression profile,

peaking at postnatal day 8 (PND8) while displaying significantly lower levels at both postnatal day 1 (PND1) and after postnatal day 20 (PND20), consistent with previously reported enzyme activity patterns (11).

We then contextualized Pla2g4e expression profile by juxtaposing the expression pattern of other molecular markers in the same postnatal period. Specifically, in the same extracts, we quantified the NeuN and synapsin I protein content, which serve as markers for mature neurons and synapses, respectively. We found that the highest levels of Pla2g4e correspond to the early instauration of a mature neuronal phenotype, with NeuN reaching a plateau after PND8 (Fig 1A). On the other hand, the expression patterns of synapsin I and Pla2g4e appeared to be inversely related, as the synaptic marker showed a continuous increase throughout the time course and particularly after PND15, when Pla2g4e expression starts decreasing. These results suggest that Pla2g4e is involved in the stages of neuronal development that precede synapse maturation, including dendritic branching and synaptogenesis. We further assessed the cellular and spatial distribution of Pla2g4e during postnatal brain development using immunostaining on tissue sections obtained from mice at PND8, PND15, and PND30 (Fig 1B). We found that Pla2g4e is widely expressed across the brain and that, in line with Western blot analysis, its highest expression levels appear at PND8. As expected, Pla2g4e shows a mainly cytosolic expression, in cortical and hippocampal neurons, where it is particularly appreciable in the cells of the polymorphic layer of the dentate gyrus (hilus). More interestingly, the subcellular localization of this enzyme concentrates also at the proximal region of the dendrites and axon. In particular, Pla2g4e appears to localize specifically to the axon initial segment (AIS) and to cortical and hippocampal neurons at PND8, PND15, and PND30. This subcellular localization was further confirmed by double staining with ankyrin G in the molecular layer of the hippocampus at PND15 (Fig 1C). Thus, based on the expression pattern, the localization of Pla2g4e in the axonal and dendritic compartments suggests that this enzyme is involved in the formation and/or stabilization of neuronal projections. Taken together, these data suggest that Pla2g4e expression is finely regulated to guarantee proper brain development.

### PLA2G4E expression is regulated during neuronal maturation

We then decided to dissect the molecular role of Pla2g4e during neuronal development taking advantage of primary cortical and hippocampal cultured neurons. We started by quantifying Pla2g4e protein expression levels at different days in vitro (DIV) in murine primary cortical neurons, obtaining an expression pattern that resembled the curve obtained in postnatal brain tissue extracts. As illustrated in Fig 2A, Pla2g4e is undetectable at DIV1 and subsequently increases until reaching its expression peak at DIV10 after which it decreases to lower levels. Interestingly, the peak of Pla2g4e expression aligns with one of the early stages of neuronal maturation, identified as Stage 5 (25, 26), further supporting the hypothesis that Pla2g4e potentially contributes to neuronal morphogenesis.

In addition, we compared Pla2g4e expression with other markers associated with different stages of neuronal development, namely, phosphorylated cAMP response element-binding protein (pCREB)

**A**

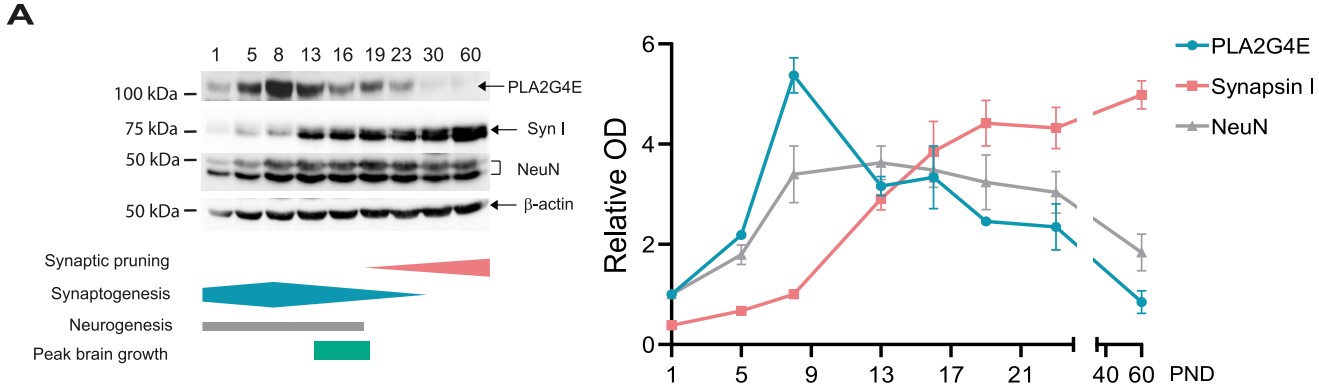

**B**

| | Hippocampus | | Frontal cortex |
| --- | --- | --- | --- |

PND8 — CA3, DG, Hilus / CA1 / Ctx

PND15 — CA3, DG, Hilus / CA1 / Ctx

PND30 — CA3, DG, Hilus / CA1 / Ctx

**C**

PLA2G4E  AnkG  Merged

and the small GTPase ADP-ribosylation factor 6 (ARF6). According to prior studies ([27]), the highest content in pCREB was detected at DIV1 after which it reached a plateau (DIV3-10), during which it contributes to the establishment of the initial neuronal dendritic complexity before sharply declining (Fig 2A). On the other hand, ARF6, which is a dendritic spine regulator in both developing and mature neurons ([28]), showed a minimal expression at DIV1 that strongly raised in the following week, reaching a sustained level that gradually increased and persisted into later stages of neuronal maturation. In this context, the peak expression of Pla2g4e corresponds to the pCREB plateau and to the instauration of elevated levels of ARF6.

This description suggests that abundant levels of Pla2g4e are crucial for the development of primary neuronal projections, whereas the establishment of a more mature neuronal profile does not require the same amount of this enzyme.

## Aberrant gene expression in neurons lacking Pla2g4e during developmental stages

To gain a broader understanding of Pla2g4e function in neuronal development, we decided to knock down its expression in primary neuronal cultures just before the peak of expression. To accomplish this, we designed three short hairpin RNA (shRNA) constructs (sh1, sh2, and sh3) against different regions of the murine *Pla2g4e* mRNA sequence (GenBank accession no. NM_001363091.1). The efficacy of the candidate shRNAs was evaluated by cotransfecting SH-SY5Y cells with two plasmids: one expressing the murine *Pla2g4e* gene pRK5-*Pla2g4e*, kindly ceded by Dr. Cravatt ([24]), and one of the three shRNA constructs, respectively.

As the sh2 construct demonstrated the best silencing effect on *Pla2g4e* protein levels (Fig S1A), we used its sequence to generate both the neuron-directed EGFP-reporter plasmid and the lentivirus for the subsequent experiments. Mouse primary neuronal cultures were transduced with either control (shCtrl) or anti-Pla2g4e (shPla2g4e) shRNA-expressing lentiviruses at DIV4, just before *Pla2g4e* expression peak. Neurons were collected at DIV14, and Western blot analysis confirmed that treatment with sh*Pla2g4e*-expressing lentiviruses significantly reduced endogenous Pla2g4e protein levels compared with the shCtrl condition (Fig S1B).

To gain a comprehensive view of the transcriptomic alterations resulting from *Pla2g4e* expression inhibition, we conducted an RNA-seq analysis of the Ctrl and sh*Pla2g4e* neuronal culture lentiviruses transduced at DIV4 and collected at DIV14. This analysis identified 1,292 differentially expressed genes (DEGs), of which 817 were down-regulated and 475 were up-regulated (false discovery rate [FDR] < 0.01) (Fig 2B). RNA-seq data generated in this study are in the GEO database under the accession number GSE59325,

subseries GSE259324. Before conducting further analyses, we confirmed a significant down-regulation in the expression of *Pla2g4e* and *Slc17a7* (which encodes the vesicular glutamate transporter 1, VGLUT1) using RT–PCR (Fig 2C). *Slc17a7* was one of the most significantly down-regulated genes after transduction with the sh*Pla2g4e* lentivirus. Interestingly, we also find that all the top 10 altered protein-coding genes have been previously associated with neuronal structure (e.g., *Tubb3* and *Neb*), activity (e.g., *Slc17a7*), or regulation (e.g., *Cck*) (Table S1), suggesting that *Pla2g4e* expression affects several neuronal functions.

Thus, to identify functional correlations among the genes affected by *Pla2g4e* silencing, we exploited gene set enrichment analysis of Gene Ontology (GO) terms to identify functional associations of DEGs. The analysis based on GO Biological Processes (BP) identified 290 significantly dysregulated pathways (FDR < 0.01). Most of these processes were predicted to be down-regulated (279 GOBP) and mainly related to neurogenesis and neuronal development in terms of both projection and cytoskeleton organization, microtubule-based processes, and synaptic signaling (Fig 2D).

To have a genome-wide overview, we also performed Reactome overrepresentation analysis (http://Reactome.org) of the down-regulated DEGs. Consistently with the GO analysis, the four "top-level" terms (Neuronal System, Vesicle-Mediated Transport, Cell–Cell Communication, and Organelle Biogenesis and Maintenance) and nine "second-level" pathways, including Nervous System Development and Signaling by Rho GTPases, imply an alteration of both neuronal structure and function in cultures lacking Pla2g4e (Fig S2A).

As these pathways are potentially associated with synapses, we subsequently conducted a synapse gene enrichment analysis using SynGO on the same set of down-regulated DEGs. Out of the 817 down-regulated DEGs, 165 corresponded to SynGO annotated genes. This analysis revealed enrichment in 21 GO Cellular Component terms and 30 Biological Process terms, including synaptic organization, signaling, and chemical transmission, which were significantly enriched (FDR < 0.01) (Fig S2B).

Finally, we analyzed our RNA-seq data considering the Human Phenotype Ontology (HPO). We found that the transcriptomic profile of neurons lacking *Pla2g4e* significantly corresponds to terms such as "Autistic Behavior" (Normalized Enrichment Score, NES): −2.23, adj. p: $3.3 \times 10^{-17}$, "Ventriculomegaly" (NES: −2.06, adj. p: $3.69 \times 10^{-14}$), "Atrophy Degeneration Affecting the CNS" (NES: −1.95, adj. p: $3.75 \times 10^{-14}$), "Upper Motor Neuron Dysfunction" (NES: −1.70; adj. p: $2.63 \times 10^{-10}$), and "Severe Intellectual Disability" (NES: −2.03, adj. p: $8.3 \times 10^{-10}$) (Fig 2E).

Taken together, the transcriptomic analysis data provide support for the hypothesis that suppressing the expression of *Pla2g4e* may impair neuronal development in terms of morphology and synaptic

**Figure 1. Pla2g4e expression is regulated throughout brain development.**
**(A)** Representative Western blot and quantification of Pla2g4e, synapsin I, and NeuN expression during the first postnatal month. Pla2g4e peaks at PND8, concurrently with synaptogenesis, and decreases during synapse maturation, when synapsin I increases (n = 3). **(B)** In hippocampus and cortex, Pla2g4e changes its subcellular localization during the postnatal period: at PND8, it is expressed in the cytosol, at the proximal region of dendrites (white arrows), and at the axon initial segment (AIS) (black arrowheads), whereas later, at PND15 and PND30, it is mainly at the AIS (black arrowheads). Scale bar, 50 μm. **(C)** In neurons of the stratum lacunosum-moleculare, Pla2g4e is expressed in the cytosol and colocalizes with ankyrin G at the AIS (white arrowheads). PND, postnatal day; DG, dentate gyrus; CA1, cornu ammonis 1; Ctx, cortex; AIS, axon initial segment; AnkG, ankyrin G. Scale bar, 50 μm.

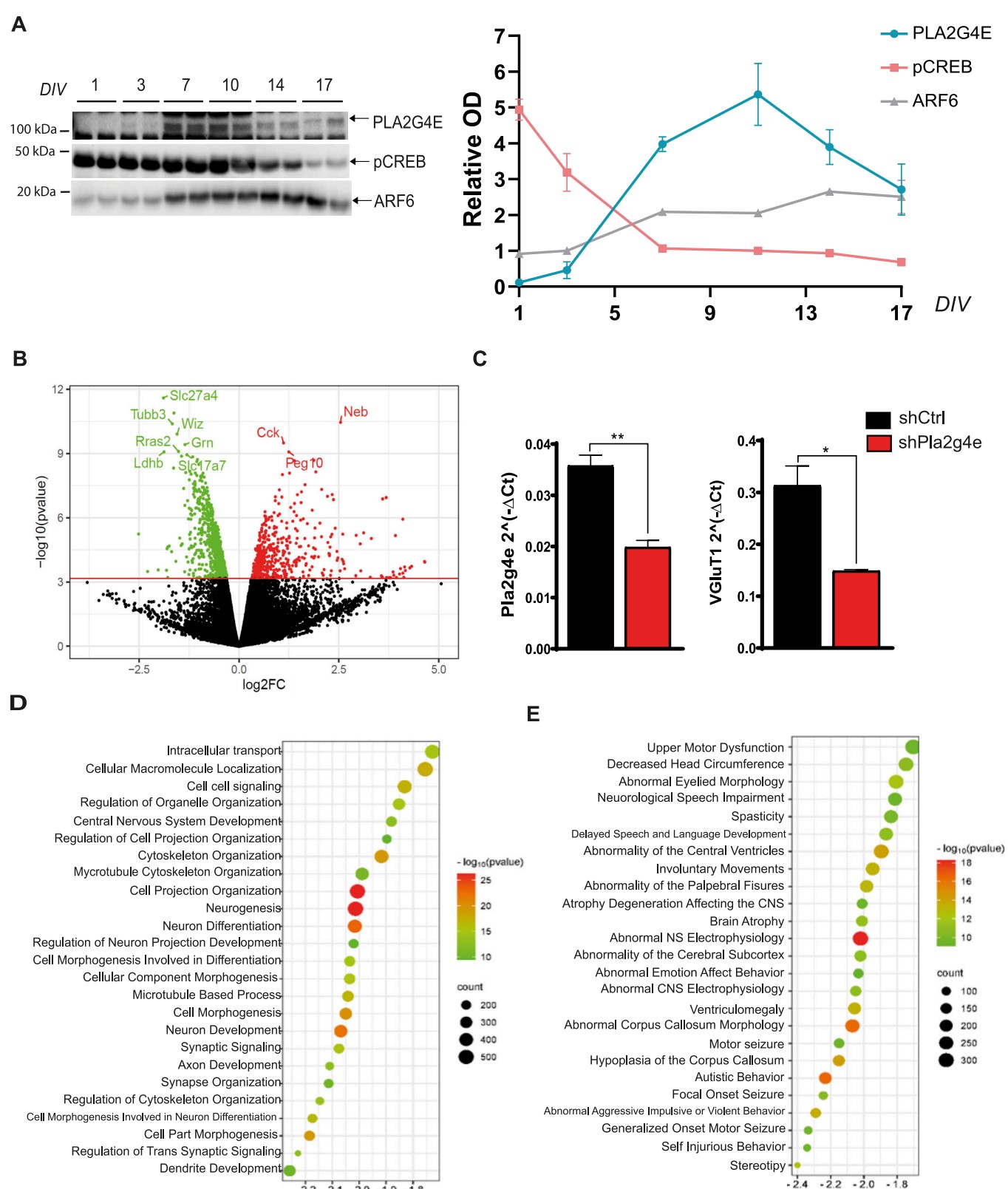

**Figure 2. RNA-seq analysis revealed neuronal transcriptomic alterations resulting from *Pla2g4e* expression inhibition.**
**(A)** Representative Western blot and quantification of Pla2g4e, pCREB, and ARF6 in mouse primary neuronal cultures during the first 17 DIV. Pla2g4e expression peaks between DIV4 and DIV14, during stage 5 of early neuron development (n = 3 per treatment). **(B)** Volcano plot of the DEGs after lentivirus-mediated *Pla2g4e* knockdown (n = 3 per treatment). **(C)** RT–PCR confirms lentivirus-mediated Pla2g4e and VGLUT1 knockdown (n = 3–4 per treatment). Data are shown as the mean ± SEM. *P < 0.05; **P < 0.01.

functionality. Moreover, these changes are reminiscent of patterns observed in certain neurodegenerative and neurodevelopmental disorders.

### Loss of *Pla2g4e* during developmental stages decreases neuronal branching in hippocampal neurons

To validate these predictions, we started by evaluating the effects of *Pla2g4e* deletion on neuronal and dendritic spine development. To achieve this, we transfected rat primary hippocampal neurons at DIV4 with a plasmid expressing EGFP under the control of a neuron-specific promoter and either scrambled (SCR, shCtrl) or the anti-*Pla2g4e*-sh2RNA construct mentioned above (sh*Pla2g4e*). At DIV14, when stable networks are formed and synaptic maturation reaches a plateau (29), neurons were fixed to perform the Sholl analysis and evaluate neuronal morphology (Fig 3A). The quantitative evaluation of the dendritic intersections at 25 to 170 μm from the soma revealed a significant reduction of the dendritic arbor complexity in neurons expressing lower levels of *Pla2g4ePla2g4e* (N = 21, P < 0.05) (Fig 3B). This result was also accompanied by the marked decrease in total dendrite length (Fig 3C). Next, we focused on dendritic spines, which serve as the ultimate recipients of functional synaptic signaling. The predominant effect of *Pla2g4e* knockdown was a notable and statistically significant decrease in dendritic spine density (N = 18–20, P < 0.0001, Fig 3D), indicating that this enzyme is required for spine formation in developing neurons. These imaging results were furthermore supported by qRT-PCR data showing a significant reduction of BDNF mRNA levels (related to dendritic arborization) and PSD-95 mRNA (related to spine density) in neuronal cultures transduced with sh*Pla2g4e*-expressing lentiviruses (Fig S2C).

Next, to examine the impact of inhibiting *Pla2g4e* expression on synaptic transmission, we recorded mEPSCs in hippocampal neuronal cultures treated with sh*Pla2g4e* and shCtrl. Knockdown of *Pla2g4e* resulted in a significant reduction of both the frequency (P = 0.007) and amplitude (P = 0.012) of mEPSCs, indicating a decrease in the release of neurotransmitter and reduced activation of postsynaptic receptors in *Pla2g4e*-lacking neurons (Fig 3E).

Finally, to confirm that the alterations detected in neurons transduced with anti-*Pla2g4e*-shRNA are strictly related to the timing of *Pla2g4e* peak expression, we transfected neuronal cultures with either the shCtrl or the sh2*Pla2g4e* EGFP-expressing plasmid at DIV10. Neurons were fixed at DIV14 to perform the Sholl analysis and dendritic length quantification (Fig 3F). Both analyses revealed nonsignificant differences between the two experimental conditions (Fig 3G and H), confirming that the expression of *Pla2g4e* is specifically required during a critical window of neuronal development.

### Altered NAPE and NAE profiles in Pla2g4e knockout mice

Subsequently, we used in vivo models to comprehensively understand *Pla2g4e* role in brain development and its influence on cognitive functions. In this sense, we took advantage of the Pla2g4e$^{tm2a}$ knockout (*Pla2g4e*$^{-/-}$ KO) mouse to assess whether the absence of Pla2g4e can lead to a neuropathological phenotype. We started validating the depletion of *Pla2g4e* protein expression by Western blot analysis of hippocampal extracts obtained from *Pla2g4e*$^{-/-}$ KO mice and WT littermates euthanized at different postnatal days. Contrarily with respect to samples from WT mice, no Pla2g4e expression was observed in hippocampal extracts from *Pla2g4e*$^{-/-}$ mice (Fig S3A). Similarly, we observed specific Pla2g4e immunostaining exclusively in brain sections from WT mice (Fig S3B).

To map the brain distribution of *Pla2g4e*, we then took advantage of the genetic design of the *Pla2g4e*$^{tm2a}$ transgenic mouse (Fig 4A), which exploits the insertion of the β-galactosidase gene (LacZ) into exon 2 of the *Pla2g4e* gene to induce the transcript decay. Thus, using X-gal staining we were able to confirm *Pla2g4e* distribution in brain mice and estimate its expression requirements using LacZ activity as a proxy. As illustrated in Fig 4B, β-galactosidase activity was evident throughout all the brain, although being more pronounced in specific areas. In detail, within the cortex the highest LacZ activity was detected in the pyriform and entorhinal cortex, and in the primary somatosensory cortex. Similarly, in the hippocampus, CA3 and hilus exhibited notably strong signals. In addition, robust activity was also detected in the *substantia nigra* in the midbrain and the granular layer in the cerebellum.

As Pla2g4e catalyzes the production of NAPEs, the precursors of NAEs, we next evaluated the endogenous levels of these lipids in the somatosensory cortex of WT and *Pla2g4e* knockout mice by LC-MS/MS. Accordingly with previous studies, basal levels of both considered classes of phospholipid were detectable in Pla2g4e$^{-/-}$ mice (11). However, certain NAPE (Fig 4C) and NAE (Fig 4D) species exhibited significant decreases in the transgenic mice. Specifically, among the 79 NAPE species considered, nine were identified in the samples, being three of them significantly reduced (P < 0.05) in Pla2g4e$^{-/-}$ mice compared with their WT littermates (PE 38:4-N-16:0, PE 36:2-N-16:0, and PE 34:2-N-16:0). Regarding NAEs and endocannabinoid species, seven out of the nine analyzed were detected. Notably, 4 NAEs (PEA, LEA, OEA, and SEA) and the endocannabinoid 2-AG showed significant decreases in Pla2g4e KO mice compared with their WT counterparts. Importantly, four of these molecules (PEA, OEA, SEA, and 2-AG) are well recognized not only as anti-inflammatory mediators but also for their neuroprotective and memory-enhancing properties (20, 30, 31, 32). Notably, the other two endocannabinoid molecules, DHEA and AEA, acknowledged for their role in synaptic modulation, showed a decreasing trend that did not reach statistical significance, likely because of the limited abundance of these species. Altogether, these results underscore

---

**(D)** Bubble plot of the top 25 Biological Process (BP) pathways and Human Phenotype Ontology terms affected by *Pla2g4e* knockdown. **(E)** Top 25 enriched HPO terms relate to neurodegenerative disorders and neurodevelopmental alterations. NES, normalized enrichment score; colors are proportional to statistical significance (minus logarithm of adjusted *P*-value); size is proportional to the gene count.
Source data are available for this figure.

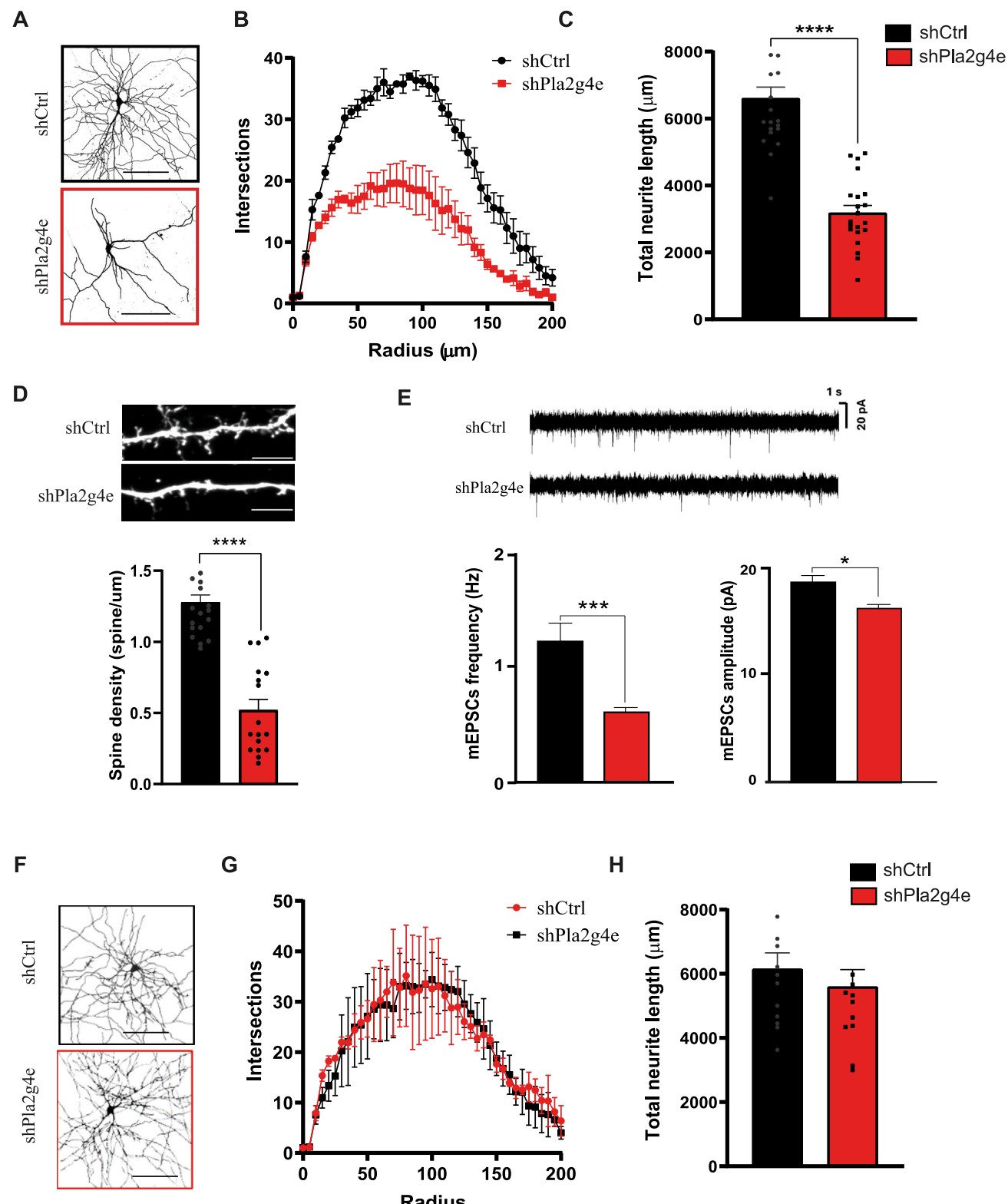

**Figure 3. Pla2g4e expression is required during developmental stages of neuronal maturation.**
**(A)** Representative images of a rat hippocampal neuron transfected with a plasmid control (shCtrl) or a plasmid containing an shRNA against Pla2g4e (sh2Pla2g4e) at day in vitro 4 (DIV4) and fixed at DIV14. Scale bar, 100 μm. **(B, C)** Sholl analysis reveals that inhibiting *Pla2g4e* at DIV4 results in decreased projection arborization and neurite length at DIV14 (n = 21 per condition). **(D)** Knockdown of *Pla2g4e* at DIV4 leads to a reduction in spine density at DIV14 (n = 18–20 per condition). Scale bar, 5 $\mu$m. **(E)** Knockdown of *Pla2g4e* at DIV4 impairs electrical proprieties at DIV14. Representative traces and quantitative analysis of mEPSCs recorded in sh*Pla2g4e*-transfected

that the absence of Pla2g4e disrupts the lipid homeostasis in the mouse brain.

## Pla2g4e plays a crucial role in maintaining proper cognitive and behavioral function

We then characterized the behavioral phenotype of 4–5 mo *Pla2g4e* KO mice by performing a battery of well-established tests that assess a variety of cognitive functions that are altered in neurodevelopmental and/or neurodegenerative diseases.

In motor tests (rotarod, pole, and open-field test), *Pla2g4e* KO mice exhibited no impairment compared with age-matched WT mice (Fig S4A–C). However, in the open-field test, *Pla2g4e* KO mice spent significantly more time in the center of the arena compared with WT littermates (Fig S4C). In addition, *Pla2g4e* KO mice performed similar to WT mice in the nest building and marble burying tests, suggesting the absence of an anxious phenotype (Fig S4D and E). Next, we decided to evaluate potential autistic traits, applying the social interaction test, where no difference was detected (Fig S4F). Nevertheless, *Pla2g4e* KO mice spend significantly more time than WT littermates in self-grooming (Fig 5A), a recognized indicator of repetitive behavior. Finally, memory function was assessed by means of two different tasks, the novel object recognition and the fear conditioning test. Specifically, mice lacking Pla2g4e showed a marked reduction in recognition memory (Fig 5B) and a significantly poorer associative memory starting from 48 h after the training session (Fig 5C), suggesting that Pla2g4e is involved in memory consolidation.

Taken together, these data indicate that the depletion of Pla2g4e does not affect the motor learning and performance of the animals, but gives rise to a murine model displaying neurodevelopmental traits. In particular, the reduced anxious-like behavior and the increased self-grooming suggest that $Pla2g4e^{-/-}$ resembles phenotypes commonly observed in transgenic models of intellectual disability and autism spectrum disorders. Moreover, the marked memory deficit of these mice, while being a sign of the cognitive impairment attributed to these conditions, may also share features with AD.

This hypothesis was further supported by RNA-sequencing analysis of the prefrontal cortex of WT and $Pla2g4e^{-/-}$ mice (Gene Expression Omnibus [GEO] accession number GSE259325, subseries GSE259323). Indeed, gene set enrichment analysis (GSEA) demonstrated that the transcriptomic alterations occurring in *Pla2g4e* KO mice affect Biological Processes related to cytoskeletal dynamics (e.g., "Axonogenesis," and "Regulation of Actin Filament-Based Processes"), neuronal development (e.g., "CNS Neuron Differentiation"), and synaptic transmission (e.g., "Synapse Organization" and "Neurotransmitter Secretion") (Fig S5A). Notably, GSEA also identified that the most highly enriched gene sets in $Pla2g4e^{-/-}$ mice overlapped with those known to be down-regulated in AD (Fig S5B; BLALOCK_-Alzheimer_Disease_DN is a gene set consisting of genes that are down-regulated in the brains of patients with Alzheimer's disease (33)).

## Pla2g4e overexpression is sufficient to restore memory deficits in Pla2g4e KO mice

Given these parallels, and based on our previous work demonstrating that AAV-mediated Pla2g4e overexpression restores cognitive function in an AD model (3), we next examined whether the overexpression of Pla2g4e in adulthood could reverse the behavioral deficits observed in constitutive $Pla2g4e^{-/-}$ mice. To achieve this, we employed the adeno-associated virus AAV9P31, a capsid variant known for its ability to penetrate the blood–brain barrier in mice (34), encoding the Hu*PLA2G4E* gene driven by the human synapsin I promoter.

The vector was systemically administered to $Pla2g4e^{-/-}$ mice via retro-orbital vein injection ($2.5 \times 10^{11}$ vg/Kg). 1 mo after the injection, memory function was assessed using the fear conditioning test. In this test, $Pla2g4e^{-/-}$ mice overexpressing Hu*PLA2G4E* exhibited a significant improvement in memory (Fig 5D). We confirmed the association between cognitive enhancement and AAV-derived *PLA2G4E* expression by performing RT–PCR to detect human *PLA2G4E* mRNA in the hippocampus (Fig 5E), as well as Western blot analysis of PLA2G4E protein levels in both cortical and hippocampal extracts (Fig 5F). In addition, IHQ targeting the HA tag, fused to the transgene, revealed widespread HA expression across various cortical regions and the hippocampus (Fig 5G). These results suggest that a diffuse neuronal overexpression of PLA2G4E can effectively restore normal memory when cognitive deficits are already established, highlighting the therapeutic applicability of the AAV9P31-Hu*PLA2G4E* viral vector in neuropathological conditions even beyond the pure AD.

## Impact of Pla2g4e deficiency on synaptic function and development

In line with the observed learning and memory impairments in $Pla2g4e^{-/-}$ mice, we analyzed the levels of key synaptic markers in the hippocampus and prefrontal cortex of adult animals. However, using Western blot analysis, no significant differences were detected in the expression of any of the pre- and postsynaptic markers considered (namely, synapsin I, VGLUT1, PSD95, and GluA1) (Fig S6A and B). Similarly, the density of dendritic spines on secondary dendrites of CA1 pyramidal neurons did not differ between $Pla2g4e^{-/-}$ and WT mice at 2–3 mo of age (Fig S6C).

Next, we investigated hippocampal CA3-CA1 synaptic contacts to assess potential alterations in synaptic electrophysiological function. We first analyzed basal synaptic transmission by applying isolated stimuli of increasing intensity to the Schaffer collaterals (Fig S6D). For a range of stimulation intensities, the slopes of $Pla2g4e^{-/-}$ fEPSP responses were not significantly different from the fEPSP responses of WT slices (n = 6 slices for WT versus 7 slices for $Pla2g4e^{-/-}$, at least three mice, Fig S6D). To further explore synaptic plasticity, because activity-dependent synaptic plasticity including short-term plasticity and long-term potentiation (LTP) alters neural

---

cultures and shCtrl (n = 24). **(F)** Representative images of a neuron transfected with shCtrl or sh2*Pla2g4e* at DIV10 and fixed at DIV14. Scale bar, 100 μm. **(G, H)** Sholl analysis shows that *Pla2g4e* knockdown after its peak expression does not have an impact on projection arborization (n = 13 per condition). *$P < 0.05$, ***$P < 0.001$, ****$P < 0.0001$.

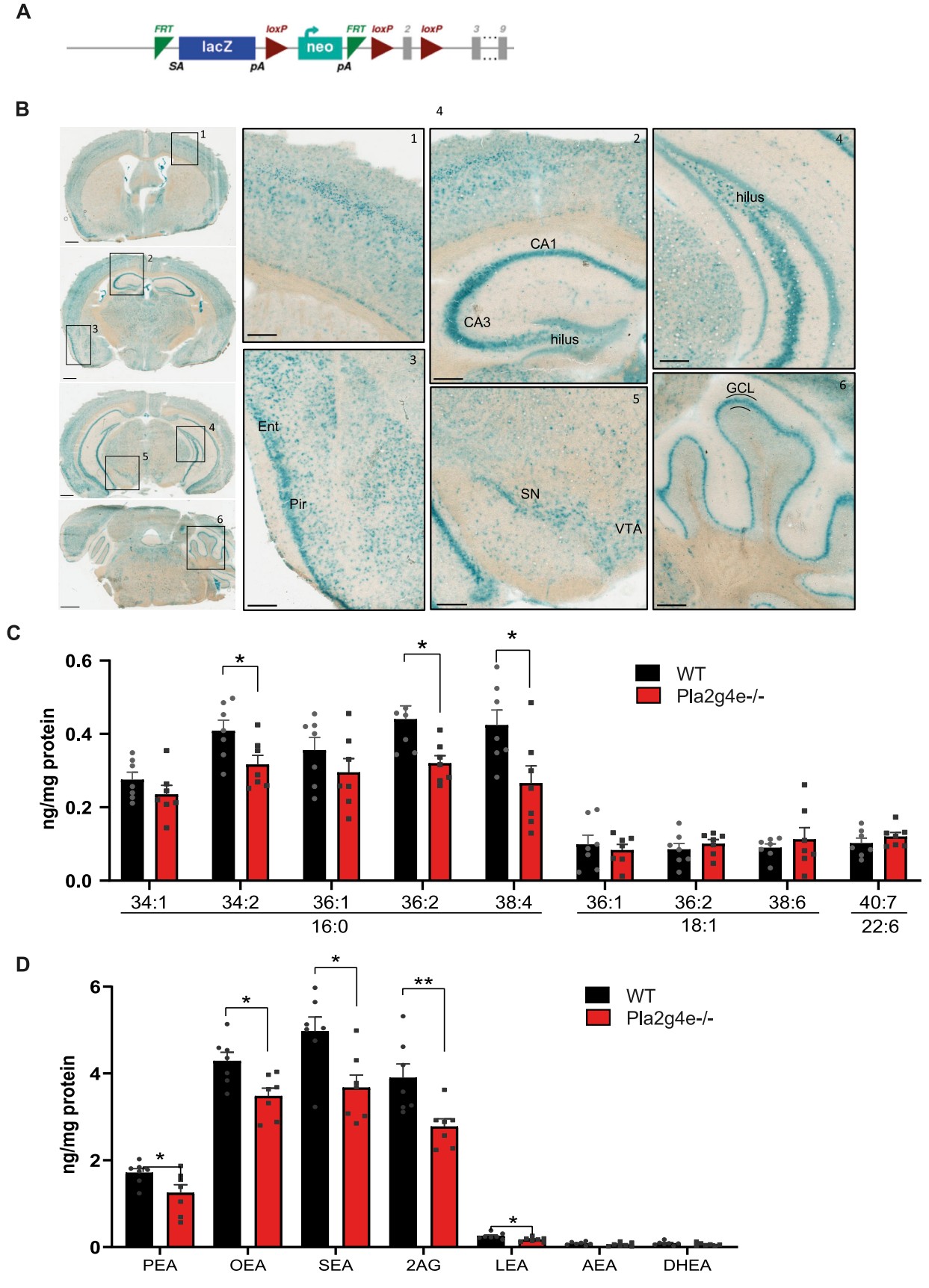

circuitry in response to recent activity, we next study the paired-pulse facilitation (PPF), as a form of short-term plasticity (35) and theta-burst conditioning tetanus to induce classical LTP (35). PPF ratios of fEPSP slopes at interstimulus intervals ranging from 25 to 325 ms were normal in the $Pla2g4e^{-/-}$ mice (n = 6 slices for WT versus 7 slices for $Pla2g4e^{-/-}$, at least three mice, at least three mice, Fig S6E). In addition, theta-burst conditioning revealed a similar ability to support LTP in $Pla2g4e^{-/-}$ and WT mice (n = 6 slices for WT versus 7 slices for $Pla2g4e^{-/-}$, at least three mice, Fig S6F). Collectively, these results indicate that Pla2g4e deficiency does not affect synaptic function at either the pre- or postsynaptic levels. Therefore, the behavioral alterations observed in $Pla2g4e^{-/-}$ mice are unlikely to be attributed to changes in adult hippocampal synaptic function.

Given that the expression of Pla2g4e is specifically required during a precise stage of brain development, we aimed to analyze the developmental profile of various synaptic marker proteins in the cortex and hippocampus of both WT and $Pla2g4e^{-/-}$ mice. This analysis was conducted from postnatal day (PND) 1 to 30. Line graphs show that in the cortex of WT mice, all the neuronal markers analyzed exhibit a significant increase in expression levels from PND1 to PND30. In contrast, $Pla2g4e^{-/-}$ mice exhibited no significant increase in expression for most markers, with the exception of synapsin and synaptophysin, which are synaptic vesicle markers (Fig 6A and B). A similar analysis in hippocampal extracts revealed fewer differences between genotypes, as developmental changes in synaptic marker expression were also observed in $Pla2g4e^{-/-}$ mice (Fig S7), suggesting that hippocampal synaptogenesis is less dependent on Pla2g4e.

Interestingly, direct comparisons between WT and $Pla2g4e^{-/-}$ mice at the specific time points showed significant reductions in key synaptic proteins, including GluA1, GluR2/3, PSD95, and GAD67, in the cortex of knockout mice, specifically at PND9 (Fig 6A and B). This observation is consistent with the peak in Pla2g4e expression at this developmental stage and with its higher expression levels in the cortex compared with the hippocampus (Fig S8). These results suggest that high levels of Pla2g4e contribute to the expression of key components of functional postsynaptic density complexes. Moreover, its absence seems to have a more significant impact on the brain development in cortical areas. Thus, aiming to detect potential hippocampal alterations, we analyzed crude synaptosomes isolated from PND10 pups using flow cytometry. Crude synaptosomes were stained with two antibodies recognizing Nrx1$\beta$ and GluA1, a pre- and a postsynaptic marker, respectively, that are expected to be co-expressed in structurally intact synaptosome units. Using flow cytometry, we found that the percentage of double-positive (Nrx1$\beta^+$/ GluA1$^+$) events is significantly decreased in the hippocampal extracts obtained from $Pla2g4e^{-/-}$ mice with respect to WT littermates (Fig 6C), suggesting that the lack of Pla2g4e results in an abnormal structure of synapses in the early postnatal period.

Taken together, these results demonstrate that the absence of Pla2g4e disrupts the developmental expression of synaptic markers in both the cortex and hippocampus, particularly during the first postnatal weeks. This delayed maturation of functional synapses likely contributes to the cognitive deficits observed in $Pla2g4e^{-/-}$ mice.

## Cognitive alterations in tamoxifen-inducible forebrain-specific *Pla2g4e* KO aged mice

Next, we investigated whether Pla2g4e expression during adulthood is required for maintaining optimal cognitive function during aging. To address this question, we first analyzed Pla2g4e expression levels in the prefrontal cortex at various time points across the lifespan. Notably, as depicted in Fig 7A, expression levels exhibited a pronounced increase at 9 mo of age, followed by a subsequent decline as the animal continues to age. This biphasic expression pattern aligns with data reported by the Mouse Dementia Network (mouseac.org). Although the underlying mechanisms regulating this dynamic expression remain unclear, we hypothesize that Pla2g4e activity is particularly important during critical and vulnerable periods such as neurodevelopment and aging. Accordingly, in order to determine the functional significance of this up-regulation, we employed a conditional knockout model to target *Pla2g4e* in CAMKII-positive excitatory pyramidal neurons ($Pla2g4e^{lox}$/lox; CAMKCreER$^{T2+/-}$). Specifically, *Pla2g4e* depletion was induced by administering tamoxifen to mice aged 7–8 mo on five alternate days and the fear conditioning and the NOR tests were conducted when the animals reached 14 mo of age (see scheme in Fig 7B).

Unlike the constitutive *Pla2g4e* KO mice, which displayed marked impairment of fear memory at 14 mo of age, the age-matched inducible conditional *Pla2g4e* KO mice exhibited normal fear memory 24 h after the training phase (Fig 7C). It is important to note that in this case, the freezing behavior of the constitutive *Pla2g4e* KO mice corresponds to 14-mo-old animals, a considerably older cohort compared with the 4- to 5-mo-old mice examined in Fig 5. At this advanced age, cognitive decline is more pronounced, likely because of the combined effects of aging and the absence of *Pla2g4e*. This age difference should be carefully considered when interpreting and comparing the behavioral outcomes presented in the two figures. Furthermore, the total absence of freezing behavior may be also related to the way the system records movement. It is important to note that in this context of motion tracking using a fear conditioning system, mobility is quantified over predefined time intervals (e.g., every 5 s). If the immobility value recorded for a given interval is zero, this does not necessarily mean that the animal was continuously moving throughout the entire period. It is possible that the animal remained still for a few seconds (e.g., 2 s), but did not meet the minimum duration threshold required by the system

**Figure 4. Characterization of Pla2g4e knockout/LacZ knock-in mice ($Pla2g4e^{-/-}$).**
**(A)** Schematic representation of the knockout allele construction. **(B)** X-gal staining highlighting the regions where *Pla2g4e* is mostly expressed. Ent, entorhinal cortex; Pir, piriform area; CA1/CA3, CA1, cornu ammonis 1/3 of the hippocampus; SN, substantia nigra; VTA, ventral tegmental area; GCL, granular cellular layer of the cerebellum. Scale bar, 800 $\mu m$ (left), 400 $\mu m$ (right). **(C, D)** NAPE and NAE contents are reduced in the parietal cortex of $Pla2g4e^{-/-}$ mice with respect to WT littermates. (C16:0, palmitic acid; C18:1, oleic acid; and C22:6, docosahexaenoic acid or DHA) (n = 5–6 per group). Data are shown as the mean ± SEM. *$P$ < 0.05.

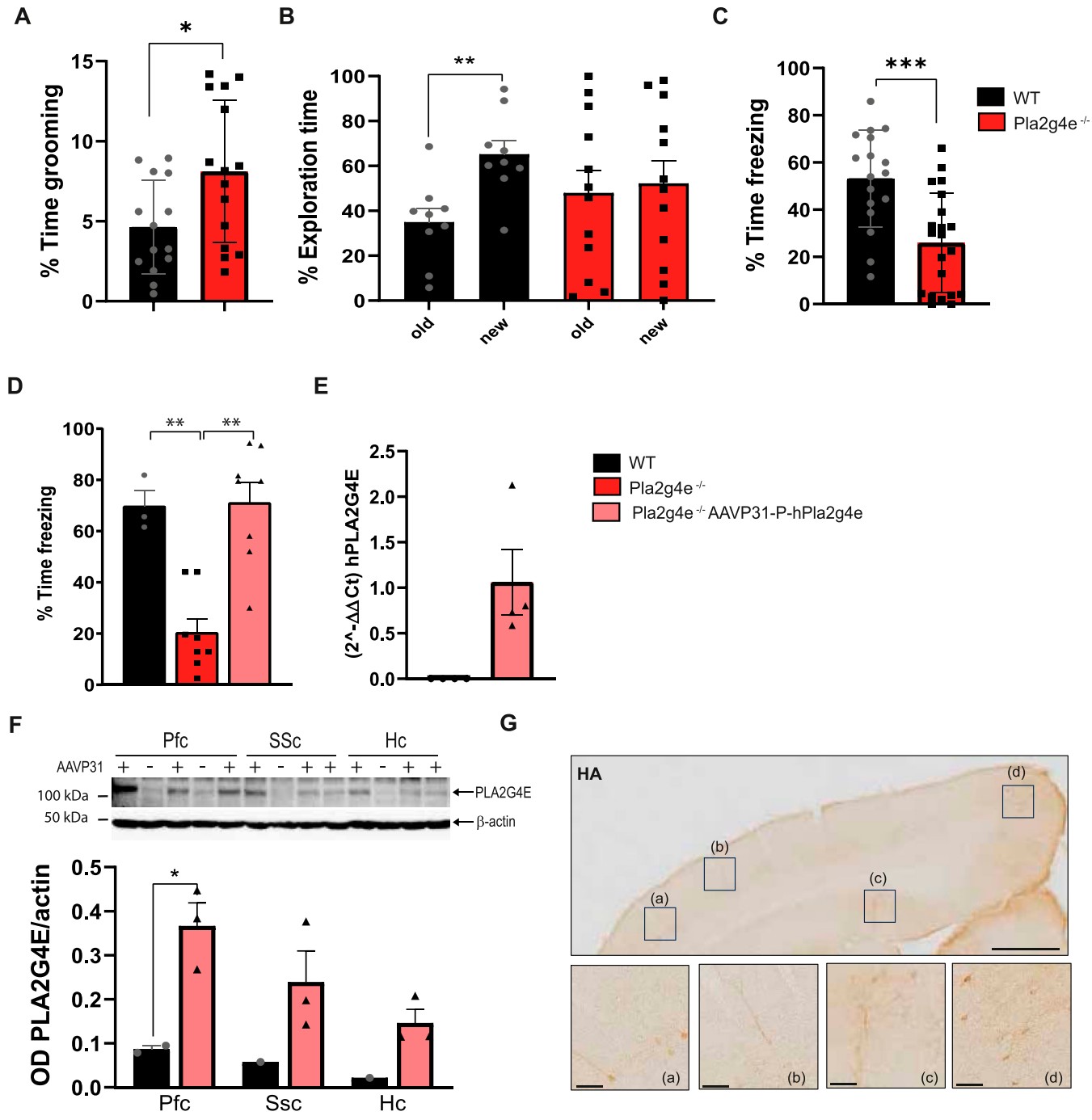

**Figure 5. PLA2G4E plays a crucial role in maintaining proper cognitive and behavioral function.**
**(A)** $Pla2g4e^{-/-}$ mice show an increased grooming behavior in comparison with WT mice (n = 14–15 per group). **(B)** Novel object recognition test reveals impaired recognition memory in $Pla2g4e^{-/-}$ mice compared with the WT group. **(C)** $Pla2g4e^{-/-}$ mice show a significantly reduced associative memory in the fear conditioning test compared with WT mice (n = 9–15 per group). **(D)** Fear memory was rescued upon AAVP31-mediated human $PLA2G4E$ expression in the brain of adult $Pla2g4e^{-/-}$ mice (n = 8 per group). **(E)** AAV9P31-mediated human $PLA2G4E$ overexpression (2.5 × $10^{11}$ vg/Kg) is detectable in the hippocampus by RT–PCR (n = 4 per group). **(F)** AAV9P31-mediated human $PLA2G4E$ overexpression (2.5 × $10^{11}$ vg/Kg) is detectable 2 mo after treatment in protein extracts obtained from the prefrontal cortex (PFC), the somatosensory cortex (SSC), and the hippocampus (Hc) of treated mice (n = 4 per group). **(G)** Representative HA staining in cortical sections of Pla2g4e$^{-/-}$ mice injected with AAV9P31-HA-PLA2G4E delivery. Scale bars, 500 μm (top image), 50 μm (bottom images). Data are shown as the mean ± SEM.*$P$ < 0.05, **$P$ < 0.01, ***$P$ < 0.001, ****$P$ < 0.0001. Source data are available for this figure.

for the episode to be registered as immobility. Therefore, zero immobility values may still include brief moments of stillness that fall below the system's detection criteria.

In contrast to fear memory assessment, in the NOR test, both the constitutive and the conditional $Pla2g4e$ KO mice demonstrated a similar poor performance, which was significantly worse than that

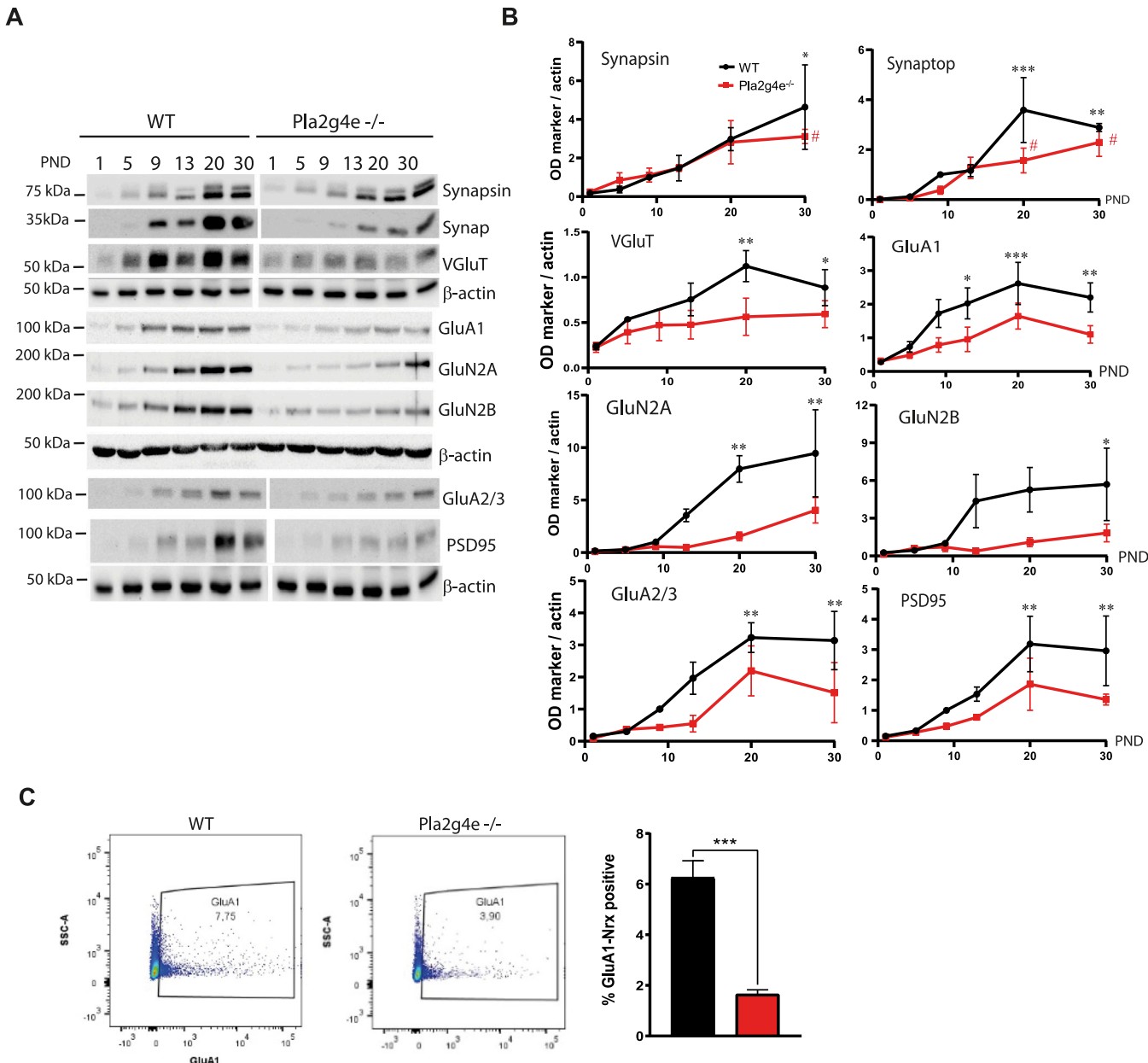

**Figure 6. Pla2g4e KO mice exhibit decreased expression of synaptic markers.**
**(A, B)** Representative Western blot and quantification of the expression of different synaptic proteins in mouse cortex along the first postnatal month (n = 3 mice per group and time point). Two-way ANOVA followed by Tukey's post hoc test; *P < 0.05, **P < 0.01, ***P < 0.001. **(C)** At PND10, the percentage of hippocampal synaptosomes expressing GluA1 is significantly lower in *Pla2g4e*[−/−] mice with respect to their WT littermates (n = 4). Data are shown as the mean ± SEM. ***P < 0.001. PND, postnatal day. Source data are available for this figure.

of aged-matched controls (Fig 7D). Western blot analysis confirmed that tamoxifen administration effectively induced the complete silencing of Pla2g4e in both the hippocampus and prefrontal cortex, ensuring that the memory tests were indeed conducted in the absence of PLA2G4E (Fig 7E). This finding indicates that the absence of Pla2g4e in fully developed adult brains partially affects cognitive performance during aging.

These results highlight that Pla2g4e expression is critically important during specific periods of heightened brain vulnerability, including both neurodevelopmental stages and neurodegenerative processes.

## Discussion

In this study, we have investigated the molecular mechanisms associated with Pla2g4e, exploring its role in cognitive processes

Figure 7. **Tamoxifen-inducible forebrain-specific conditional knockout mice lacking *Pla2g4e* exhibited preserved memory function.**
**(A)** *Pla2g4e* transcript levels are significantly increased in the hippocampus of 9-mo-old mice (n = 4). **(B)** Schematic representation of the schedule of tamoxifen administration and behavioral test carried out with *Pla2g4e*[lox/lox] CAMKCreER[T2+/−] mice. **(C, D)** *Pla2g4e* knockout in excitatory neurons at 7–8 mo of age does not affect 14-mo-old mouse fear memory (C) but alters the recognition memory (D) in the same conditional *Pla2g4e* KO mice (n = 9–14 per group). **(E)** Western blot analysis confirms the lack of *Pla2g4e* expression in the hippocampus and prefrontal cortex of tamoxifen-induced conditional *Pla2g4e* KO mice. Data are shown as the mean ± SEM. **$P < 0.01$, ***$P < 0.001$, ****$P < 0.0001$. TMX, tamoxifen; NOR, novel object recognition; Hc, hippocampus; PFC, prefrontal cortex.

and assessing its importance for maintaining optimal brain function. Our research reveals that the expression of neuronal Pla2g4e reaches its highest levels during the early postnatal days, a crucial period for brain development. Interestingly, when we selectively down-regulated Pla2g4e expression in neuronal cultures during early neurodevelopmental stages, coinciding with peak synaptogenesis,

we observed a reduction in dendritic arborization and spine density, ultimately impacting neuronal function in vitro. We found that constitutive Pla2g4e knockout mice exhibited altered developmental profile of synaptic plasticity markers, leading to relevant behavioral alterations in adulthood, particularly in memory function. Remarkably, these deficits were effectively corrected by restoring the

expression of Pla2g4e in neurons. In addition, selectively removing Pla2g4e from excitatory neurons in the mature brain partially impacted memory function in aged animals, suggesting a crucial role of Pla2g4e in cognitive aging. Our study altogether suggests that Pla2g4e plays a crucial role in orchestrating synaptogenesis during brain development and in cognitive aging. Furthermore, it provides additional support for the efficacy of a Pla2g4e-based gene therapy in neurological disorders characterized by synaptic degeneration.

The bell-shaped protein expression pattern of Pla2g4e observed during mouse brain development aligns with previous studies, indicating that the *Pla2g4e* transcript peaks in the neonatal mouse brain, with the highest enzymatic activity found at PND7 (11, 24). Pla2g4e is the enzyme responsible for producing neuroprotective NAPEs and NAEs during ischemic events in neonatal brains (11). In physiological conditions, NAPEs represent only ~0.01% of total phospholipids in mammalian cells (36) and their increase is required only under situations of injury or stress, such as ischemia. In these circumstances, the elevation in $Ca^{2+}$ levels accelerates the synthesis of NAPEs, thereby facilitating the subsequent formation of NAEs, which accumulate and conceivably exert a protective function (12, 37). Thus, in their post-decapitation ischemia model, Rahman and colleagues detected a more-than-100-fold increase in 18 NAE species, including N-stearoylethanolamide (SEA/18:0 NAE, 208-fold), N-palmitoylethanolamide (PEA/16:0 NAE, ~392-fold), and N-oleoylethanolamide (OEA/18:1 NAE, ~104-fold), whose formation was strongly impaired in *Pla2g4e*$^{-/-}$ KO mice. These data suggest that NAPEs, NAEs, and consequently Pla2g4e could play a neuroprotective role in the first developmental stages of the CNS. Neonates generally exhibit heightened susceptibility to hypoxia-induced ischemic insults compared with adults (11). Consequently, the observed up-regulation of Pla2g4e expression at an early postnatal time point may be associated with neuroprotective mechanisms.

Interestingly, this time window, where higher levels of Pla2g4e expression are recorded (approximately between PND5 and PND8), has been described as a critical postnatal period for brain development, particularly in terms of brain growth and synaptogenesis (38, 39). Thus, it could be reasonable to consider Pla2g4e as a main effector in synaptogenesis, hypothesis that is furtherly supported by previous interventional studies that identified the increase in dendritic spine density as one of the main Pla2g4e-mediated beneficial processes in different mouse models (3, 4).

Interestingly, thanks to immunohistochemical analysis we could also detect a peculiar subcellular localization of Pla2g4e in the AIS, which was furtherly confirmed by the colocalization with ankyrin G in the hippocampus. AIS is a highly specialized neuronal compartment where action potential is initiated (40). In this sense, Pla2g4e could be involved in the instauration and maintenance of this structure for proper connectivity and function. On the other hand, we could detect neuronal populations that exhibit an intense staining at PND8, including mossy cells in the dentate gyrus. Notably, these cells contribute to the proper functioning of the hippocampus and accurate performance in fear conditioning tests (41). In addition, high expression levels were detected in the motor and primary somatosensory cortex previously reported by reference 11. Thus, these observations indicate a widespread presence of Pla2g4e in most of the neuronal types, suggesting that it could play a role in several essential neuronal processes.

Accordingly, the highest expression of Pla2g4e in neuronal cultures corresponds to a specific stage in neuronal development, namely, Stage 5 (26). This stage marks the initiation of primary projections that established the first synaptic contacts between dendritic spines and axon terminals, thereby establishing the foundational framework of a neuronal network (25). Interestingly, a comparison of protein levels revealed that Pla2g4e peaks concomitantly with the plateau of ARF6, a small GTPase implicated in synaptogenesis in both developing and mature neurons (28). In this context, the decline in Pla2g4e levels after PND15 suggests that in contrast to ARF6, this enzyme is primarily essential during the early stages of neuronal development. This hypothesis is further supported by the Sholl analysis of neurons expressing Pla2g4e at optimal levels until DIV10, which exhibited normal dendritic arborization. In line with this, we demonstrated that the inhibition of Pla2g4e during the early stages of neuronal development exerts a substantial impact on their transcriptomic profile in the mature state. Of particular interest, *Slc27a4* emerged as the most down-regulated gene, encoding the long-chain fatty acid transport protein 4 (known as FATP4), a crucial protein responsible for supplying fatty acids to the fetal brain (42). This protein appears to play a critical role during neurodevelopment, a period when the brain demands a consistent and reliable supply of fatty acids. Accordingly, it has been described that alteration in fatty acid uptake within the human fetal brain, potentially influenced by the SLC27A4 dysfunction, could contribute to autism spectrum disorder pathophysiology (42). In addition, our data indicate that the inadequate expression of Pla2g4e is detrimental for VGLUT1 levels, which is essential for maintaining proper excitatory signaling. Thus, we could speculate that Pla2g4e efficiently modulates glutamatergic transmission, highlighting its pivotal role in synaptic plasticity. This hypothesis is supported by electrophysiological data from neuronal cultures, demonstrating alterations in the miniature excitatory postsynaptic currents when Pla2g4e is absent.

Importantly, beyond its enzymatic role in lipid signaling, Pla2g4e may also contribute structurally to neuronal function. Specifically, PEA and OEA, products of Pla2g4e activity, can selectively activate PPARα, thereby preventing oxidative and inflammatory processes and reducing neuronal death during cerebral injury (43). In addition, this phospholipase is known to participate in membrane trafficking, particularly within clathrin-independent endocytic and recycling pathways, which are essential for the rapid, stimulus-dependent synaptic vesicle recycling required for efficient neurotransmission. In this context, Pla2g4e may act not only through its enzymatic activity and bioactive lipid products such as NAEs, but also as a structural cargo protein that supports neuronal membrane remodeling events necessary for synapse formation and plasticity. Therefore, we cannot exclude either mechanism, or the interplay between them, as contributing factors in its role in memory formation and dendritic spine development.

Consequently, it is noteworthy the diminished expression of both pre- and postsynaptic molecules observed in the brain of the constitutive *Pla2g4e* knockout model during the developmental stage, ultimately impairing cognition in adult animals. In this context, constitutive Pla2g4e$^{-/-}$ mice exhibited significantly impaired associative and recognition memory. Furthermore, an AD-like phenotype was predicted in the RNA-seq comparing WT and

Pla2g4e KO brain tissue. Furthermore, the transcriptomic profile of neuronal cultures lacking *Pla2g4e* significantly correlated not only with AD-related phenotypes but also with conditions such as "Autistic Behavior" and "Severe Intellectual Disability," terms that were surprisingly in accordance with the altered self-grooming phenotype recorded in vivo. Fascinatingly, these conditions are linked to neurodevelopmental alterations that align well with the unique pattern of Pla2g4e expression. Accordingly, diverse regulation of this gene bears implications for both neurodevelopment and neurodegenerative disorders. Human observational studies have noted differentially methylated CpGs within the *PLA2G4E* gene between preterm infants with an optimal neurobehavioral profile compared to infants with a poorly regulated neurobehavioral profile ([23]). Furthermore, genetic variants within the *PLA2G4E* gene have been suggested as plausible risk factors for neurodevelopmental issues, notably including panic disorder ([22]). Collectively, these findings substantiate the idea of a pivotal role played by this enzyme in the phase of neurodevelopment, contributing to adult brain health.

In the conditional *Pla2g4e*[lox/lox];CAMKCreER[T2+/−] knockout model, where proper brain neurodevelopment occurred, the absence of Pla2g4e during adulthood results in a less severe impact, affecting only specific types of memory, particularly, impaired performance in the object recognition task but preserved fear conditioning. This dissociation likely reflects the distinct neural substrates underlying these tasks: object recognition is primarily dependent on cortical regions such as the perirhinal and prefrontal cortex, whereas fear conditioning relies more on the amygdala and hippocampus. Notably, synaptic marker alterations in the *Pla2g4e* KO mice were more pronounced in the cortex, suggesting that Pla2g4e plays a critical role in maintaining synaptic integrity in cortical circuits required for object-based memory, but is less essential for subcortical circuits mediating associative fear learning.

This finding suggests that PLA2G4E's significance extends beyond brain development to include at least partial involvement in cognitive aging. Future studies should investigate whether critical conditions, such as the accumulation of age-related changes or exacerbation of Alzheimer's disease neuropathology, could hasten the onset of dementia in the absence of PLA2G4E.

It should be noted that the restoration of a normal fear memory in the complete Pla2g4e KO, while serving as a proof of concept that the reported alterations are directly related to the lack of Pla2g4e, presented an important opportunity to test a novel BBB-permeable capsid. Importantly, the results demonstrated that the AAV9P31 capsid ([34]), despite being administered intravenously, can efficiently enter the brain displaying even distribution and physiological expression, while still eliciting cognitive improvements. In conclusion, we propose that PLA2G4E is a fundamental regulator of neuronal development that plays a distinct and noncompensable function. Accordingly, PLA2G4E should be adequately expressed particularly in critical contexts such as infancy, brain injury, and aging, wherein its protective role can be effectively exerted.

Our study suggests that besides the physiological contribution of PLA2G4E during brain development and aging, this investigation will be useful for the translation of *PLA2G4E*-based gene therapy from in vivo testing to clinical applications. Specifically, we propose that the timing and target diseases for its application could be more effectively tailored, thereby augmenting the likelihood of successful outcomes and broadening potential clinical applications.

# Materials and Methods

### Mice

All the procedures were conducted in accordance with the European and Spanish regulations (2010/63/EU; RDS52/2013) and were approved by the Ethical Committee for the Animal Experimentation of the University of Navarra (protocols: 082c-21 (*Pla2g4e*[tm2a]), 112c-21 (*Pla2g4e*[lox/lox]), 015-22 (*Pla2g4e*[tm2c], *Pla2g4e*[lox/lox], *Pla2g4e*[lox/lox] CAMKCreER[T2+/−])). Mice were bred and housed in the animal facility of the Centre for Applied Medical Research (CIMA) in Pamplona. Mice were housed 3–6 per cage with free access to food and water and maintained in a 12-h light–dark cycle in controlled temperature and humidity conditions. Behavioral tests were conducted between 9:00 and 17:00. Both males and females were included in the experimental procedures.

### Constitutive and inducible conditional *Pla2g4e*[−/−] mice

The constitutive *Pla2g4e* knockout C57BL/6N-A[tm1Brd] *Pla2g4e*[tm2a] (*EUCOMM, Hmgu/BayMmucd*, RRID: MMRRC_037767-UCD) mouse was obtained from the Mutant Mouse Resource and Research Centre (MMRRC). In these transgenic mice, the *Pla2g4e*[tm2a] allele is predicted to generate a null allele through splicing to a LacZ trapping element. The conditional knockout *Pla2g4e*[tm2c] was then generated by removal of the gene-trap cassette by FLP recombinase, which produced the floxed *Pla2g4e* gene. *Pla2g4e*[lox/lox] mice were crossed with heterozygous CaMKCreER[T2] mice ([44]) to obtain a time-specific knockout of *Pla2g4e* gene in excitatory neurons. Tamoxifen (T5648; Sigma-Aldrich) was dissolved at a final concentration of 20 mg/ml by shaking overnight (o/n) at 37°C in a rotary mixer. Both experimental (*Pla2g4e*[lox/lox] CAMKCreER[T2+/−]) and control (*Pla2g4e*[loxt/lox] CAMKCreER[T2−/−]) adult mice were administered via oral gavage during five alternate days with a 75-mg/kg dose of tamoxifen. Behavioral tests were performed 1 mo from the last administration.

#### Behavioral tests

To minimize bias, all behavioral experiments were conducted with the experimenter blinded to the genotype of the animals. In addition, because preliminary analyses revealed no significant behavioral differences between male and female *Pla2g4e* KO mice, both sexes were included and used interchangeably to ensure generalizability.

#### Pole and rotarod test

In the pole test, mice were placed on top of the pole with the head oriented upward and the time required for the animals to orient themselves downward and to descend to the base was recorded in three trials. The rotarod test was performed taking advantage of an accelerating rotarod (Harvard Apparatus) where in the two first days, mice were trained to coordinate and resist on the rotarod at

least respectively 30 and 60 s (training phase), and in the third day, latency time before falling was recorded in one trial.

### Fear conditioning test

Fear conditioning (FC) test was performed using a StartFear system (Panlab). In the first day, mice were put in the conditioning chamber with no stimuli and allowed to freely explore for 2 min (habituation phase). The day after, animals were placed in the same chamber and received two footshocks (0.2 mA) of 2 s after 90 and 120 s, respectively, and were returned to their home cages after an additional 30 s (training phase). Mouse memory was estimated considering the percentage of time spent "freezing" during the test consisting in 2 min in the conditioning chamber without any stimuli. The test was performed 24 h, 48 h, 1 wk, or 12 d post-training.

### Open-field and novel object recognition test

Open-field (OF) and novel object recognition (NOR) tests were performed in a modular open-field arena (45 × 45 × 45) with opaque walls, and mouse behavior was recorded with an Action Camera (National Geographic) using the video tracking system Smart 3.0 (Panlab). In the first day, mice were allowed to freely explore the arena for 5 min. In this session, motor behavior was recorded and analyzed in terms of total distance and time in the center (OF). The day after, two identical objects were located in the same arena in parallel position, and mice could freely explore them for 5 min (training phase). The day after, recognition memory was tested by changing one of the two objects and considering the time that mice spent interacting with the novel and the familiar object. Results were expressed as the percentage of exploration time around the novel or the old object.

### Social interaction test

Social approach test was performed following the previously published protocol (45) to assess social behavior. Data were expressed as the difference between the time in the interaction zone (TI) during the target session ($TI_T$) ant the TI in the no-target session ($TI_{NT}$). The "target" animal was changed every 4–5 sessions to reduce the deriving stress.

### Self-grooming

Spontaneous grooming behavior was evaluated using the previously described protocol (46). Briefly, each experimental mouse was individually placed in a clean, empty mouse cage without bedding. After a 10-min habituation period, the animals were rated for 10 min for cumulative time spent grooming all body regions.

### Marble burying and nest building tests

Mable burying and nesting tests were performed following the previously described protocols (47). For the marble burying test, standard polycarbonate rat cages (26 cm × 48 cm × 20 cm) were filled with fresh, unscented mouse bedding material to a depth of 5 cm and 12 standard glass toy marbles (assorted styles and colors, 15 mm diameter, 5.2$g$ in weight) were placed on the surface of the bedding (3 columns x 4 rows). Each mouse was allowed to freely move in the cage for 30 min after which the number of unburied marbles was quantified. For nest building test, mice were singularly placed in a standard polycarbonate rat cage box where a piece of

tightly packed cotton material (Nestlets Nesting Material, Ancare) had been added. Mice were left free to interact with the material o/n, and the day after, nesting index was evaluated and scored as no nest, partial nest, or complete nest covering the mouse.

### Animal euthanasia and tissue fixation

For biochemical studies, mice were euthanized by cervical dislocation and brains were dissected at 4°C. The prefrontal cortex (Fig S9A), hippocampus (Fig S9B), and somatosensory cortex (Fig S9C) were carefully isolated as illustrated and rapidly frozen in dry ice and stored at -80°C until use. For X-gal staining, fresh brains extracted from cervical dislocated animals were rapidly frozen with isopentane in dry ice. Coronal and sagittal sections of 10- to 30-$\mu$m thickness obtained with a cryostat were stored on Epredia SuperFrost Plus Adhesion slides (Thermo Fisher Scientific) at –20°C until use. For Golgi–Cox staining, fresh hemi-brains extracted from cervical dislocated animals were directly immersed in Golgi–Cox solution that, after 48 h, was replaced with the same, fresh solution where brain tissue remained for three more weeks before processing as indicated below.

For immunohistochemistry, after being anaesthetized with an intraperitoneal (i.p.) injection of a combination of ketamine and xylazine (80/10 mg/kg), animals were perfused with PBS for 3 min followed by 12 min with 4% PFA. Brains were immersed in 4% PFA for 24 h at RT before being changed to EtOH 70% (PanReac) for 3 d at RT before being included in paraffin blocks.

### Protein extracts and immunoblotting

The tissue was homogenized in a lysis buffer containing 2% SDS, 10 mM Tris–HCl (pH 7.5), phosphatase inhibitors (1 mM NaF and 0.1 mM $Na_3VO_4$), and 1x Complete Protease Inhibitor Cocktail (Roche). The sample was finally sonicated and centrifuged at 16,000$g$ to recover the supernatant. Samples were mixed with a 6X Laemmli sample buffer and resolved onto SDS–polyacrylamide gels and transferred to a nitrocellulose membrane using the Trans-Blot Turbo Transfer System. Membranes were blocked with 5% milk in TBS for 1 h at RT and incubated o/n at 4°C with the corresponding primary antibody at the dilution shown in Table S2. After three washes in TBS/Tween-20, membranes were incubated with HRP-conjugated anti-rabbit or anti-mouse antibody (1:5,000, Santa Cruz Biotechnology) for 1 h. Antibody binding to specific protein bands was visualized via an enhanced chemiluminescence system (ECL, GE Healthcare Bioscience, or Pierce ECL Plus, Thermo Fisher Scientific). Images were acquired and quantified using ImageLab Software (Bio-Rad).

### Immunohistochemistry and immunofluorescence

Immunohistochemistry (IHC) and immunofluorescence (IF) were performed in paraffin slides. In detail, the tissue was deparaffinized and treated with either citrate buffer (pH 6.0) (IHC) or Tris-EDTA buffer (pH 9.0) (IF) for antigen retrieval. Sections were stained o/n at 4°C, using single or combined primary antibodies at the dilutions indicated in Table S2. For IHC, after washing in PBS, sections were incubated with biotinylated goat anti-rabbit secondary antibody.

A specific signal was visualized using an avidin–biotin–peroxidase complex with 3,3'-diaminobenzidine tetrahydrochloride (DAB) as the chromogen (diluted 1:500; Invitrogen). Hematoxylin IHC counterstaining was performed for nucleus visualization. Images were obtained using the automated system ScanScope (Aperio). For IF staining, after the o/n incubation with primary antibodies, slices were washed in PBS before adding Alexa Fluor 555– or Alexa Fluor 647–conjugated secondary antibodies before being coverslipped in mounting medium with DAPI. Colocalization images were obtained on a confocal microscopy (Zeiss LSM 800; Zeiss).

### X-gal staining

$\beta$-Galactosidase activity because of LacZ expression was detected by X-gal staining of cryostat brain sections. Tissue was first fixed by incubation in a 0.5% glutaraldehyde/PBS solution for 10 min at RT. After washing with PBS, slices were incubated o/n at 37°C in a humid chamber with a solution consisting of 5 mM $K_3Fe(CN)_6$, 5 mM $K_4Fe(CN)_6$, 1 mM $MgCl_2$, and 1 mg/ml X-gal in PBS. The day after, slides were washed with PBS and mounted with aqueous Epredia Shandon Immu-Mount (Thermo Fisher Scientific) medium. Images were obtained with ScanScope (Aperio).

### Phospholipid extraction and analysis

On average, 30 mg of parietal cortex tissue was homogenized in 250 $\mu$l methanol: acetonitrile (1:1) using TissueLyser (QIAGEN) for 2 min. Samples were then centrifuged for 5 min at 20,000$g$ at 4°C. The supernatant was collected and furtherly centrifuged for another 5 min at 20,000$g$ at 4°C, whereas the pellet was kept at –20°C for protein quantification. The supernatant resulting from the second centrifuge was filtered in a deep well plate, and solvent was evaporated using nitrogen gas flow. The dry pellet was resuspended in 100 $\mu$l of methanol. Phosphatidylethanolamine (PE) was used as an internal standard. 10 $\mu$g of each sample was separated and analyzed using an HPLC-MS/MS system consisting of an HPLC Binary SL 1200 series (Agilent Technologies) column (XTerra RP18 3.5 $\mu$m 4.6 x 100 mm column (Waters) mobile phase: 100% MeOH, 0.1% formic acid, isocratic, 0.500 $\mu$l/min) and a PAL HTS-xt autosampler connected to an AB Sciex API 4000 Triple Quad LC/MS/MS. We used a positive electrospray ionization (ESI) source and a triple quadrupole analyzer. Data were obtained using the software Analyst. For normalizing data, the pellet obtained from the first centrifuge was resuspended in 50 $\mu$l RIPA buffer and protein content was quantified with the Pierce BCA Assay kit (Thermo Fisher Scientific). NAPE and NAE contents were expressed as the ratio between lipid (ng) and protein (mg) content.

### Fluorescence analysis of single synapse (FASS)

Synaptosome analysis was performed as previously described ([48], [49]). Briefly, crude synaptosomes were obtained from freshly dissected mouse hippocampi. Brain tissues were homogenized using a glass/Teflon homogenizer in a sucrose buffer (320 mM sucrose, 10 mM Hepes, pH 7.4, protease inhibitors [1:1,000, #P8340; Sigma-Aldrich], and phosphatase inhibitors [1:100, #78420; Thermo Fisher Scientific]) and centrifuged at 1,200$g$ for 10 min to remove cell debris. The supernatant (S1) was further centrifuged at 12,000$g$ for 20 min to obtain the synaptosome-enriched pellet (P2). Synaptosomes were resuspended by gently pipetting up and down in 1.5 ml of extracellular solution (120 mM NaCl, 3 mM KCl, 2 mM $CaCl_2$, 2 mM $MgCl_2$, 15 mM glucose, and 15 mM Hepes [pH 7.4]). After blocking with 4 ml of ice-cold blocking buffer (5% FBS in PBS), resuspended synaptosomes were incubated for 30 min at 4°C with primary antibodies against GluA1 (#13185, 1:1,500; Cell Signaling) and Nrx1$\beta$ (75-216, 1:400; NeuroMab), and calcein AM (100 nM, #65-0853-39; eBioscience). The samples were then washed with blocking solution, 2,500$g$ for 10 min at 4°C. Pelleted synaptosomes were resuspended and incubated for 30 min at 4°C with secondary antibodies anti-rabbit Brilliant Violet 421 (#111-675-144, 1:400; Jackson ImmunoResearch) or anti-mouse Alexa Fluor 647 (A21240, 1:800; Invitrogen), and calcein AM (100 nM) in the dark. Synaptosomes were washed again and resuspended in 500 $\mu$l PBS and stored protected from light at 4°C until they were analyzed by flow cytometry.

Samples were acquired using FACSCanto II System (BD Biosciences) using the protocol previously set up in reference [49]. Synaptosome integrity was confirmed by calcein-associated fluorescence, and a total of 10,000 gated events were recorded. Data were analyzed using BD FACSAria Fusion cytometer and FlowJo software (LLC) ([49]). Synaptosomes were selected in the SSC versus FSC graph (size ~0.75 to 3 $\mu$m). Doublets and multiplets were discriminated based on forward scatter height (FSC-H) versus forward scatter area (FSC-A) plots and excluded from subsequent analyses. Specific immunostaining was determined by gating particles following standard immunostaining protocols for flow cytometry using fluorescence minus one (FMO) samples (respectively no anti-GluA1 and no anti-Nrx1$\beta$). Double-immunostained synaptosomes are represented as the percentage of the total detected synaptosomes.

### Dendritic spine density measurement by Golgi–Cox staining

The half-brains were soaked in a Golgi–Cox solution (1% potassium dichromate, 1% mercury chloride, 0.8% potassium chromate) for 3 wk, and the previously described protocol was followed ([50]). Spine density was determined in the secondary apical dendrites of CA1 hippocampal pyramidal cells arising at distances from the soma of between 100 and 200 $\mu$m. Average data were obtained by quantifying the spine density in three dendrites of three neurons of three brain slices obtained from each brain (n = 27 dendrites for each group).

### Ex vivo electrophysiology

Detailed methods of most of the procedures have been described previously ([35], [51]). Briefly, coronal slices (350 $\mu$M thickness) containing the hippocampal formation were made from 3-mo-old mice using a vibratome (Vibratome Series 3000 Plus) and incubated for 1 h at RT in artificial cerebrospinal fluid (ACSF) gassed with 95% $O_2$, 5% $CO_2$, pH 7.3. The ACSF contained 124 mM NaCl, 2.69 mM KCl, 1.25 mM $KH_2PO_4$, 2 mM $MgSO_4$, 26 mM $NaHCO_3$, 2 mM $CaCl_2$, and 10 mM glucose. Slices were then transferred to an immersion recording chamber and superfused at 2 ml/min at 30 ± 2°C. Extracellular field

excitatory postsynaptic potentials (fEPSPs) were recorded with a carbon fiber microelectrode placed in the stratum radiatum of CA1 and evoked by stimulation of the Schaffer collateral fibers with an extracellular bipolar tungsten electrode via a 2,100 isolated pulse stimulator. Basal synaptic transmission was analyzed at the beginning of each experiment by applying stimuli of increasing intensity to reach a maximal fEPSP response. Then, the stimulation intensity was adjusted to give fEPSP response of ~50% of maximal fEPSP and was kept constant throughout the experiment. To measure PPF, different interpulse intervals were used (25–325 ms). For LTP experiments, after recording stable baseline responses for 10 min, LTP was induced by applying four trains (1 s at 100 Hz, five bursts of five pulses, with an interval of 200 ms between bursts) spaced 30 s, and potentiation was measured for 1 h after LTP induction at 0.2 Hz. Changes in the fEPSP slope were calculated in relation to the baseline. To isolate fEPSP from GABAA-mediated inhibitory synaptic transmission, 50 $\mu$M of picrotoxin was used.

### AAV vector for in vivo administration

The AAV9P31 capsid was generated by inserting 7-mer AQWPTSYDA peptide between residues 588 and 589 in the hypervariable surface loop VIII of AAV9 as described in reference 34. The inclusion of this peptide significantly increases the capacity of AAV9 to cross the blood–brain barrier (34). The AAV9P31-SYN1-HAh*PLA2G4E* was diluted with PBS to reach the working concentration of 1 × 10$^{11}$ genome copies (gc)/$\mu$l. 60 $\mu$l of the virus, corresponding to a dose of 2.5 × 10$^{11}$ vg/Kg, was administered to each mouse by retro-orbital injection.

### Primary neuronal cultures

For RNA analysis and total extract biochemical studies, murine primary cultures of cortical and hippocampal neurons were established from E18 C57BL/6N embryos adapting the previously described protocol (52). Neurons were plated at a density of 7.5 x 10$^5$ neurons per well in 6-well plates (Costar) coated with laminin (BD Biosciences) and poly-D-lysine (Millipore).

For imaging studies, hippocampal neuronal cultures were established from E19 rat fetuses following a well-established protocol (53, 54) with minor modifications. Cells were collected or fixed at DIV14.

### Cloning of anti-Pla2g4e shRNA

Three shRNA template oligonucleotides (sh1, sh2, and sh3), based on three different segments of the *Pla2g4e* murine gene (GenBank accession no. NM_001363091.1), were designed using the siRNA Target Finder and Design Tool i-Score Designer. Three oligonucleotides encoding different *Pla2g4e*-targeting shRNAs (Table S2) were cloned into the pSUPER.neo+gfp (OligoEngine) vector for small interfering RNA (siRNA) following the manufacturer's protocol. Briefly, complementary oligonucleotides bringing HindIII and BglII ends were annealed in a 100 mM NaCl and 50 mM Hepes, pH 7.4, buffer at a final 60 $\mu$g/ml concentration. In parallel, 1 $\mu$g of pSUPER.neo+gfp plasmid was linearized with BglII and HindIII restriction enzymes. The linearized vector was purified using QIAquick Gel Extraction Kit (QIAGEN). Annealed oligonucleotides and linearized pSUPER.neo+gfp were ligated in 3:1 M ratio by overnight T4 DNA Ligase (Invitrogen) reaction at RT.

The recombinant vector was transformed in *E. coli* DH5α competent cells (Thermo Fisher Scientific) following the supplier protocol, and 100 $\mu$g/ml ampicillin (Sigma-Aldrich)-containing LB agar (CONDA) plates were used for antibiotic selection. Single bacterial colonies were grown overnight in 3.5 ml LB broth (CONDA) containing 100 $\mu$g/ml ampicillin (Sigma-Aldrich), and DNA was extracted using QIAprep Spin Miniprep Kit (QIAGEN). The presence of positive clones was checked by digestion with EcoRI and HindIII restriction enzymes in NEBuffer 2.1 (New England Biolabs). Clones producing a specific 281-bp band were finally sequenced with a short hairpin-specific Sanger sequencing protocol. The silencing efficiency of the obtained recombinant plasmids was tested in SH-SY5Y cells transfected with murine *Pla2g4e*-expressing plasmid (pRK5-*Pla2g4e*), kindly ceded by Dr. Cravatt (24), and cotransfected with the pSUPER recombinant vector expressing one of the three anti-*Pla2g4e* shRNAs.

For primary neuron treatments, shCtrl and sh2*Pla2g4e* shRNA were successively subcloned into a pLVTHM (Trono Lab) plasmid. Exploiting the EcoRI-HF and ClaI restriction enzymes, the obtained plasmids were amplified using QIAGEN Plasmid Maxi Kit (QIAGEN) and analyzed by Sanger sequencing for the following applications.

### Lentivirus production and transduction

60–70% confluent HEK293T cells were transfected with 9 $\mu$g shRNA-expressing plasmids, 6 $\mu$g psPAX2 and 3 $\mu$g pMD2. G (both from Trono Lab), using Lipofectamine 3000 (Invitrogen) system and by diluting the mixture in Opti-MEM (Gibco). Cells were maintained in a CO2 incubator at 37°C for 24 h before completely being discarded and renewing culture medium. The new medium was then collected after 24 h and substituted with fresh medium that was finally recollected the day after. The collected lentivirus-containing media were centrifuged for 10 min at 2,000*g* to remove cell debris, and the resulting 0.45-$\mu$m filtered supernatant was centrifuged at 50,000*g* at 4°C for 2.5 h in Beckman Coulter L-100XP Ultracentrifuge (SW28 bucket). The supernatant was discarded and pellet resuspended in PBS before being aliquoted in a suitable volume, quick-frozen, and stored at −80°C. Lentivirus production was tested by transducing HEK293T cells with different amounts of virus and checking for EGFP fluorescence using a Leica DM IL LED microscope (Leica).

Lentiviruses were added directly to the culture medium of primary neurons, and two thirds of the medium was replaced with fresh Neurobasal medium after 6 h. The most suitable sh*Pla2g4e* lentivirus dose was determined as the minimum amount of lentivirus needed to halve the expression of the *Pla2g4e* transcript (analyzed by real-time PCR). The same amount of lentivirus carrying an shCtrl was used for the basal experimental condition.

### Calcium phosphate transfection of primary neuronal cultures

For imaging studies, neurons were transfected with the calcium phosphate method. 30 min before transfection, complete culture medium was collected and maintained at 37°C, whereas

prewarmed MEM (or DMEM) with GlutaMAX was added to neurons. Depending on the transfection efficiency, 2 to 3 $\mu$g DNA/well was mixed with CaCl$_2$ and H$_2$O and added drop by drop to an equal volume of HeBSS2x while vortexing. After a 30-min incubation at RT, protected by light, the mixture was added drop by drop to neurons that were rapidly put back in the incubator. After 10 min, precipitate formation was confirmed with an optical microscope, and neurons were washed with DMEM-GlutaMAX before adding back to the original complete culture medium.

### Neuronal imaging studies

For neuron morphology studies, transfected hippocampal neurons were fixed 10 min at RT with PBS containing 4% PFA and 4% sucrose. After washing with PBS, neurons were permeabilized with 0.1% Triton X-100 for 15 min at RT and incubated with 5% BSA in PBS (blocking solution) for 1 h at RT. Cells were labeled with GFP primary antibody (Table S2) diluted in blocking solution for 1 h at RT. Cells were then washed and incubated with fluorophore-conjugated secondary antibodies for 1 h at RT. After washing, nuclei were stained with DAPI and mounted on glass slides with Epredia Shandon Immu-Mount (Thermo Fisher Scientific). Individual neurons were imaged using the 40x oil objective of a Zeiss Confocal LSM 510 system (Zeiss) at 0.6 zoom. Eight to ten 0.5-$\mu$l section images were acquired with a sequential acquisition setting with a 1,024 × 1,024 pixel resolution. The Sholl analysis (55) was performed manually using Fiji Simple Neurite Tracer (SNT) plugin (56). For spine density quantification, three dendrites *per* neuron were imaged with 63x oil objective of a Zeiss Confocal LSM 510 system (Zeiss) at 1.0 zoom. Spine density was expressed as the number of spines/analyzed dendrite length ($\mu$m).

### RNA isolation

RNA extraction was performed using Maxwell RSC simplyRNA Tissue Kit (Promega) following the manufacturer's protocol. Thanks to magnetic beads supplied in the Maxwell RSC Cartridge, Maxwell RSC Instrument performed RNA extraction and DNase I treatment resulting in 25-50 $\mu$l RNA-containing solution that was quantified with NanoDrop 1,000 (Thermo Fisher Scientific) before storage at −80°C until use.

### RNA-seq

After confirming RNA integrity via Agilent RNA Nano LabChips (Agilent Technologies), a total of 1 $\mu$g of the total RNA was used to construct cDNA libraries with the TruSeq Stranded mRNA Kit (Illumina). The libraries were quantified using quantitative real-time polymerase chain reaction (qRT-PCR) according to the qRT-PCR Quantification Protocol Guide. An Agilent Technologies 2100 Bioanalyzer was used for the qualification.

For RNA-sequencing data analysis, the quality of the samples was verified using FastQC software version 0.11.8 and the alignment of reads to the mouse genome (mm10) was performed using STAR software. Gene expression quantification was carried out considering read counts of exonic gene regions with featureCounts and using Gencode M18 as the gene annotation reference.

Differential expression statistical analysis was performed using R/Bioconductor. First, gene expression data were normalized with edgeR and voom and genes with read counts lower than 6 in 100% of the samples were excluded from the study. LIMMA (Linear Models for Microarray Data) was used to identify the genes with significant differential expression between experimental conditions. Further data analysis was performed using various tools: R/Bioconductor was used for clustering analysis, Gene Ontology and Reactome were employed for enrichment analysis, and additional graphical representations were performed with SRPlot and online Venny 2.0.2.

### Reverse transcription and qRT-PCR

To obtain cDNA, 2 $\mu$g of RNA was reverse-transcribed using SuperScript III Reverse Transcriptase (Invitrogen) in the presence of its buffer (Invitrogen), 10 mM dNTPs (Invitrogen), Random Primer (Invitrogen), DTT (Invitrogen), and RNaseOUT (Invitrogen). This mixture was incubated for 60 min at 37°C and 5 min at 90°C before being used or stored at −20°C.

Quantitative real-time PCR was performed to quantify the most interesting genes resulting from RNA-seq analysis. More in detail, 4 $\mu$l of the retro-transcribed cDNA (0.01–0.07 $\mu$g) was mixed with 7 $\mu$l Power SYBR Green PCR Master Mix (Applied Biosystems) and 0.5 $\mu$l of 10 $\mu$M forward primer, 0.5 $\mu$l of the corresponding 10 $\mu$M reverse primer, and water to reach a final volume of 14 $\mu$l. All the primers used are shown in Table S2. Assays were performed in triplicate in a 96-well plate, and probe amplification and analysis were carried on in QuantStudio 5 Real-Time PCR System. For the analysis, the employed reference gene was 36B4 (57) and relative gene expression was calculated using the $2^{(-\Delta(\Delta Ct))}$ method (58).

### In vitro electrophysiology

Electrophysiological experiments were performed on DIV14 hippocampal cultured neurons. Excitatory postsynaptic currents in miniature (mEPSCs) have been measured by patch-clamp recordings in the whole-cell voltage clamp modality using the Axopatch 200B amplifier and the pClamp-10 software (Axon Instruments) as in reference 59. In detail, neurons were held respectively at −70 or +10 mV in the presence of TTX 1 $\mu$M. Recordings were performed in the Krebs-Ringer/Hepes (KRH) external solution (NaCl 125 mM, KCl 5 mM, MgSO$_4$ 1.2 mM, KH$_2$PO$_4$ 1.2 mM, CaCl$_2$ 2 mM, glucose 6 mM, Hepes-NaOH, pH 7.4, 25 mM). Recording pipettes were fabricated from glass capillary using a two-stage puller (Narishige); they were filled with the intracellular solution (cesium gluconate [CsGluc] 130 mM, CsCl 8 mM, NaCl 2 mM, EGTA 4 mM, Hepes 10 mM, MgATP 4 mM, GTP 0.3 mM), and the tip resistance was 3–5 M$\Omega$. Recordings were performed at RT, and currents were sampled at 10 kHz and filtered at 2 kHz. The recorded traces have been analyzed using Clapfit-pClamp 10 software, after choosing an appropriate threshold.

### Statistical analysis

The results were processed for statistical analysis using GraphPad Prism, version 8. The normal distribution of data was checked by the Shapiro–Wilk test. An unpaired two-tailed *t* test was used to

compare two groups, and one-way ANOVA was employed for three or more groups. Two-way ANOVA with Tukey's post hoc test was employed to analyze the immunoblotting signal of protein synaptic marker during brain development from postnatal day (PND) 1 to 30 in both WT and *Pla2g4e*$^{-/-}$ mice. Mean data from the Sholl analysis of the two groups of each experiment were obtained by row statistics. Multiple *t* tests were used for grouped analysis including the Sholl analysis and NOR. The number of replicates is indicated in the corresponding figure legends. Unless otherwise indicated, results are presented as the mean ± SEM. Statistical significance was set at *$P \leq 0.05$, **$P \leq 0.01$, or ***$P \leq 0.001$.

# Data Availability

All NGS data generated have been deposited to the NCBI Gene Expression Omnibus (GEO) under the accession number GSE259325, SubSeries GSE259323 and GSE259324 (https://www.ncbi.nlm.nih.gov/geo/query/acc.cgi?acc=GSE GSE259325), whereas all the rest of data that support the findings of this study are available on request from the corresponding author [A Garcia-Osta and M Cuadrado-Tejedor].

# Supplementary Information

# Acknowledgements

This work was supported by PID2022-138285OB-I00, financiado por MCIN/AEI/10.13039/501100011033/FEDER, UE, and PID2019-104921RB-I00 financiado por MCIN/AEI/10.13039/501100011033 to A Garcia-Osta and M Cuadrado-Tejedor; CaixaImpulse: CI20_00207 to A Garcia-Osta; the GMP and Adey Foundations, Asociación de Amigos, and FENS-IBRO-PERC exchange fellowship grants to S Badesso; and grant PID2020-116327GB-I00 to ED Martín. This work was also supported partially by the Ministry of University and Research (MUR) Progetto Eccellenza (2022–2027) to the Department of Pharmacological and Biomolecular Sciences "Rodolfo Paoletti," Università degli Studi di Milano, and partially by the Italian Ministry of Health with Ricerca Corrente and "5xmille" funds to N Mitro. F Antonucci is economically supported by Italian Ministry of Research—PRIN: PROGETTI DI RICERCA DI RILEVANTE INTERESSE NAZIONALE—No. 20229XWM5L; Telethon Foundation—Seed Grant Spring 2022 RETT—Project N. 419; and Associazione Nazionale Atassia Telangiectasia (ANAT)—ANAT Research Grants 2022. This project has received funding from the Italian Ministry of University and Research (PRIN202039WMFP and PRIN2022 PNRR P2022R2E8N to E Marcello), from the Giovanni Armenise Harvard Foundation and AIRALZH ONLUS (2023 Armenise Harvard-AIRALZH Mid-Career Award in Neurodegenerative Diseases—AHA-MCA to E Marcello), and from the Italian Ministry of Enterprises and Made in Italy (PNRR—Next generation EU funding, "SEED for Innovation Patent 2.0" program, project TT_MIN23_SEED4IP2.0_07 to E Marcello).

## Author Contributions

S Badesso: data curation, investigation, methodology, and writing—review and editing.

M Perez-Gonzalez: data curation, methodology, and writing—review and editing.

S Exposito: conceptualization, data curation, formal analysis, and methodology.

M Espelosin: data curation and formal analysis.

C Cambria: conceptualization, data curation, formal analysis, and methodology.

L D'Andrea: data curation, formal analysis, investigation, and methodology.

G Imperato: data curation, formal analysis, and investigation.

P Moeini: methodology.

A Vales: methodology.

G Gonzalez-Aseguinolaza: formal analysis and methodology.

N Mitro: conceptualization, data curation, formal analysis, funding acquisition, investigation, and methodology.

F Antonucci: conceptualization, data curation, formal analysis, funding acquisition, investigation, methodology, and writing—review and editing.

ED Martín: conceptualization, data curation, formal analysis, funding acquisition, investigation, methodology, and writing—review and editing.

E Marcello: conceptualization, data curation, formal analysis, funding acquisition, investigation, and writing—review and editing.

M Cuadrado-Tejedor: conceptualization, resources, data curation, formal analysis, supervision, funding acquisition, validation, investigation, methodology, and writing—original draft, review, and editing.

A Garcia-Osta: conceptualization, resources, data curation, formal analysis, supervision, funding acquisition, validation, investigation, visualization, methodology, project administration, and writing—original draft, review, and editing.

## Conflict of Interest Statement

The authors declare that they have no conflict of interest.

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
