## [Reviewer comments · Life Science Alliance]

Life Science Alliance

Loss of PLA2G4E compromises synaptic structure and cognitive outcomes in mice

Sara Badesso, Marta Perez-Gonzalez, Sara Exposito, Maria Espelosin, Clara Cambria, Laura D'Andrea, Gabriele Imperato, Pedram Moeini, Africa Vales, Gloria Gonzalez-Aseguinolaza, Nico Mitro, Flavia Antonucci, Eduardo Martín, Elena Marcello, Mar Cuadrado-Tejedor, and Ana Garcia-Osta

DOI: <https://doi.org/10.26508/lsa.202503323>

Corresponding author(s): Ana Garcia-Osta, *Fundación para la Investigación Médica Aplicada* and Mar Cuadrado-Tejedor, *University of Navarra, CIMA, Neurosciences Division*

Review Timeline:

Submission Date:	2025-03-25
Editorial Decision:	2025-04-24
Revision Received:	2025-05-12
Editorial Decision:	2025-06-06
Revision Received:	2025-06-11
Accepted:	2025-06-23

Scientific Editor: Tim Fessenden

Transaction Report:

April 24, 2025

Re: Life Science Alliance manuscript #LSA-2025-03323-T

Prof. Ana Garcia-Osta
Fundación para la Investigación Médica Aplicada
Neuroscience
Pio XII, 55
Pamplona, Navarra 31008
SPAIN

Dear Dr. Garcia-Osta,

Thank you for submitting your manuscript entitled "PLA2G4E: Major Player during Brain Development and Cognitive Aging" to Life Science Alliance. The manuscript was assessed by expert reviewers, whose comments are appended to this letter. We invite you to submit a revised manuscript addressing the reviewers' main concerns.

As you will see, reviewers commended the intriguing findings on Pla2g4e in mice and neurons in vitro, with implications for development, behavior, and neurodegeneration. While Reviewers 1 and 2 sought extended evidence on the role of this gene in Alzheimer's Disease, Reviewer 3 noted the explicit connection to this disease is somewhat weak. A revised manuscript should either strengthen this connection, with helpful suggestions by Reviewers 1 and 2, or reduce the emphasis on this disease as suggested by Reviewer 3. In addition, multiple reviewers commented on results shown in Fig 7 that contradict prior observations.

Next, Reviewer 3 requested important validation and controls to confirm central observations made in this work: validate specificity of anti-Pla2g4e antibodies in KO mice, rescue Pla2g4e KD for results in Fig 3, and validate the extent/level of Pla2g4e expression in vivo following AAV injection. This reviewer also sought clarification of statistical tests and whether sex as a biological variable was considered for in vivo studies.

Please note that, in view of comments from Reviewer 3, LSA requires molecular weight markers on all protein gels and a clear indication of biological and experimental replicates in figure legends for all papers that have been provisionally accepted.

Thank you for this interesting contribution to Life Science Alliance. We are looking forward to receiving your revised manuscript.

Sincerely,

B. MANUSCRIPT ORGANIZATION AND FORMATTING:

Reviewer #1 (Comments to the Authors (Required)):

The manuscript demonstrates that PLA2G4E is critically required for neuronal synaptic development and cognitive function. Depletion of PLA2G4E results in aberrant synaptogenesis in neuronal cultures and cognitive impairment in mice. Notably, PLA2G4E overexpression exhibits protective effects, positioning it as a potential therapeutic target for Alzheimer's disease (AD). While the study is intriguing, there are several issues that need to be addressed prior to publication:

1. It remains unclear whether PLA2G4E expression is altered in AD model mice or in AD patients. Investigating this could underscore the role of PLA2G4E in the pathogenesis of AD.
2. The biosynthesis of N-acyl phosphatidylethanolamines (NAPEs) and N-acyl ethanolamines (NAEs) following PLA2G4E overexpression in mice has not been studied, which is critical for understanding its functional implications.
3. Evaluating the effects of PLA2G4E overexpression in aged mice or established AD model mice would be particularly valuable. Such experiments could determine if PLA2G4E overexpression can mitigate age-related synaptic dysfunction or cognitive deficits. Positive findings in these contexts would significantly enhance the translational relevance of the research. At a minimum, a discussion of this possibility is warranted.
4. The figures require correct annotations: please ensure accurate labeling of the " μm " unit in Figure 3D and include scale bars in Figures 3A, 3D, 3F, and 4B to comply with journal formatting standards.

Reviewer #2 (Comments to the Authors (Required)):

In this manuscript, the authors analyzed the functions of PLA2G4E in the brain using its knockout mice and showed that PLA2G4E is involved in memory-related behavior. This work provides new insights into the biological roles of PLA2G4E in brain development and cognitive function. However, the following points need to be clarified to strengthen the authors' conclusion.

Major comments

(1) LC-MS/MS analysis revealed a decrease in 2-AG in Pla2g4e knockout mice (Figure 4D). PLA2G4E is an N-acyltransferase that produces NAPEs and NAEs, but not 2-AG, which is produced by different pathways. What is the authors' take on this?

(2) In Figure 5D, the overexpression of PLA2G4E exhibited a significant improvement in memory. It is of interest to evaluate whether the administration of NAEs, the product of the PLA2G4E reaction, can also improve memory.

(3) PLA2G4E shows a bell-shaped expression profile, peaking at PND8 and declining thereafter (Figure 1A), while Figure 7A indicates peak expression at the later stage, 9 months. What is the difference?

(4) The biological functions of PLA2G4E are thought to be exerted through its reaction product, NAEs. PPAR α and CB1 are known to be the receptors for NAEs and may be involved in the results observed in this study. The authors should discuss the possible mechanism involving these and other receptors for NAEs.

Minor comments

The explanations of the Figures and their legends are not clear and need improvement.

There are multiple bands in NeuN in Figure 1A and PLA2G4E in Figure 2A. The authors should specify the position of target proteins.

In Figure 1C, please specify PND8, 15, and 30.

The explanation of Figure 1D is missing.

In Figure 5F, please clarify "V" and "S" above the blot.

Reviewer #3 (Comments to the Authors (Required)):

This manuscript describes studies of Pla2g4e in the mouse brain, and impacts of constitutive and conditional knockouts as well as rescue of behavioral deficits by rAAV-mediated expression by peripheral to central delivery of Pla2g4e. Overall the data package is pretty diffuse, the data pieces are somewhat disparate and though they lead to some potentially interesting insight into Pla2g4e and impacts on neurons and behavior in mice, a lot of the data lacks rigor reducing the confidence of the results. The most interesting studies on rAAV overexpression rescue and inducible knockouts lack depth in terms of characterizing the in-situ effects.

General comments. No MW markers on any gel. Extra bands suggest the antibody is not specific. No evidence for key replication of data. Sex as a biologic variate is not examined. No evidence for experimental blinding in behavioral studies

Minor note mouse Pla2g4e is being studied in most cases not human, and it should not be all caps.

The "spin" of the paper is on AD (at least the first sentence) but the paper really has little if anything to do with AD, it's really a study of Pla2g4e and behavior. The focus on AD might want to be shifted. Indeed, the discussion barely relates the findings to AD.

Overall, the paper is loosely written and hard to follow with respect to what was done. It would be improved by editing for conciseness and flow.

Specific concerns on the data and studies.

Figure 1. No data documenting the specificity of the Pla2g4e antibody is provided. No MW or full-length gels provided. If the same antibody being used as in Figure 2a this clearly has many other cross-reactive bands. This renders the IHC and IF in the figure suspect. Having two antibodies, some absorption controls, even overexpression in situ would help. Actually, the KO brains should be used as a control (from studies in Figure 4)

Figure 2. No control for toxicity in the shRNA experiments. As many studies the rescue experiment which is the best control is not performed (co-express a non-targeted Pla2g4e and show the effects reverse). The RNAseq data is shown but does RNA seq support reduction of Pla2g4e and Vglut1? If not, then one worries about false discovery. $-\log_{10}$ p values are shown not a p adjusted for multiple testing. Primary cell studies must be repeated, it's too easy to have a one off from a non-healthy culture. No assessments of toxicity are provided,

Figure 3. This is not inhibition but knockdown. Again, no assessments of off target effects or toxicity. Again, were these experiments repeated? Can the knockdown effect be rescued with Pla2g4e co-expression. No Pla2g4e staining is shown. One might predict that the cells with decreased arbors should have lower PLA2g4e expression.

Figure 5. The rAAV overexpression study is perhaps the most potentially impactful data, but appears to be highly underpowered, sex as a biologic variable is not assessed, and data on in situ overexpression is not provided. The calculation of vg appears in error, or if it is correct then the vg used are almost certainly incompatible with mediating high levels of expression in the brain. Methods and text are hard to reconcile on this calculation. The lack of in situ data showing how many cells are actually expressing Pla2g4e is a major concern. The behavioral effect looks impressive but if only few cells are getting transduced and expressing Pla2g4e a confound exists. The labeling of the western blot in figure 5 is not described. What is V what is S? What about looking at synapses or other assessments of phenotypes in the brain?

Figure 7. Again, no assessment of the phenotype of the conditional and constitutive knockouts in terms of synapses, gliosis, neuronal health etc...

Why is the FC data different for the constitutive knockout between the two studies shown (figure 7 and before)

Response to Reviewers

Life Science Alliance manuscript "PLA2G4E: Major Player during Brain Development and Cognitive Aging" #LSA-2025-03323-T

We sincerely thank the editor and reviewers for their careful reading and helpful comments. In response to the concerns raised, we have conducted additional experiments and incorporated new data and clarifications into the revised manuscript. These revisions are detailed in our point-by-point response and are highlighted in the updated version of the manuscript. We believe these changes have significantly strengthened the manuscript and hope that all the reviewers' concerns have been fully addressed.

Reviewer 3 noted the explicit connection to this disease is somewhat weak. A revised manuscript should either strengthen this connection, with helpful suggestions by Reviewers 1 and 2, or reduce the emphasis on this disease as suggested by Reviewer 3. In addition, multiple reviewers commented on results shown in Fig 7 that contradict prior observations.

Thank you very much for your comment. Although PLA2G4E was initially identified in the context of Alzheimer's disease, your observation is appropriate and constructive. We have revised the manuscript to shift the focus toward the physiological role of this phospholipase, which is the central aim of this study. Accordingly, we have reduced the emphasis on Alzheimer's disease throughout the text, as suggested. Regarding the seemingly contradictory observations, we believe they are entirely compatible. It is plausible that this enzyme exhibits a peak of expression during the postnatal period and again around 8–9 months of age followed by a slight decrease as the animal continues to age. In fact, this biphasic expression pattern is consistent with data reported for this enzyme in the Mouse Dementia Network web server (mouseac.org). To validate our results, the observation found in this web has been now included and referenced in the manuscript.

Next, Reviewer 3 requested important validation and controls to confirm central observations made in this work: validate specificity of anti-Pla2g4e antibodies in KO mice, rescue Pla2g4e KD for results in Fig 3, and validate the extent/level of Pla2g4e expression in vivo following AAV injection. This reviewer also sought clarification of statistical tests and whether sex as a biological variable was considered for in vivo studies.

The specificity of the antibody is demonstrated in Supplementary Figures S3 and in Figure S1 the silencing efficiency of anti-Pla2g4e shRNA sequences was assessed in SH-SY5Y cells co-transfected with a murine PLA2G4E-expressing plasmid (pRK5-PLA2G4E) and a pSUPER recombinant vector encoding one of the three anti-Pla2g4e shRNAs. As shown in wells 3 and 4, a distinct band (110 kDa) confirms the expression of PLA2G4E following transfection with the pRK5-PLA2G4E plasmid. The intensity of this band was notably reduced in wells corresponding to cell extracts co-transfected with the plasmid and the shRNA-expressing vectors, indicating effective knockdown. In Supplementary Figure S3A, PLA2G4E expression is detectable in wild-type mice but absent in knockout (KO) mice, further supporting the antibody's specificity. Furthermore, Figure 3B confirms the specificity of the antibody in immunohistochemistry, as the signal observed in the dentate gyrus (DG) of the hippocampus in wild-type mice is absent in knockout (KO) mice.

In the *in vivo* studies, since in our preliminary analyses no behavioral differences between male and female KO mice were observed, both sexes were used interchangeably in the experiments. Additionally, although it was missing in the methods section, all behavioral experiments were conducted with the experimenter blinded to the genotype of the animals to minimize bias. We acknowledge these comments. Both aspects have been now clarified in Methods section.

Please note that, in view of comments from Reviewer 3, LSA requires molecular weight markers on all protein gels and a clear indication of biological and experimental replicates in figure legends for all papers that have been provisionally accepted.

We have included the molecular weight markers on all protein gels and a clear indication of biological and experimental replicates in figure legends.

Reviewer #1 (Comments to the Authors (Required)):

The manuscript demonstrates that PLA2G4E is critically required for neuronal synaptic development and cognitive function. Depletion of PLA2G4E results in aberrant synaptogenesis in neuronal cultures and cognitive impairment in mice. Notably, PLA2G4E overexpression exhibits protective effects, positioning it as a potential therapeutic target for Alzheimer's disease (AD). While the study is intriguing, there are several issues that need to be addressed prior to publication:

1. It remains unclear whether PLA2G4E expression is altered in AD model mice or in AD patients. Investigating this could underscore the role of PLA2G4E in the pathogenesis of AD.

Thank you for the comment. This information is highly relevant and has been described in detail in our study by Pérez-González et al., which is cited in this manuscript. There, we reported that PLA2G4E expression is reduced in the brains of both mouse models and Alzheimer's disease (AD) patients compared to healthy controls. Although this phospholipase was initially identified in the context of AD, the primary focus of this study is its physiological role in the central nervous system. In line with this, and following the reviewer's suggestion, we have revised the text to minimize the emphasis on Alzheimer's disease.

2. The biosynthesis of N-acyl phosphatidylethanolamines (NAPEs) and N-acylethanolamines (NAEs) following PLA2G4E overexpression in mice has not been studied, which is critical for understanding its functional implications.

We appreciate the reviewer's comment. While the measurement of NAEs is indeed valuable, we believe it is not essential for the current manuscript. We have already confirmed, in independent experiments, that overexpression of PLA2G4E leads to increased levels of NAEs in the brain, and we are actively working on publishing these findings separately. In the present study, our goal was to perform a proof-of-concept rescue experiment in PLA2G4E knockout mice to demonstrate that the observed memory deficits are specifically due to the absence of PLA2G4E. It is reasonable to assume that restoring PLA2G4E expression increases its products—including NAEs. We believe that additional quantification of NAEs, while supportive, is not critical for the main conclusion of this work.

3. Evaluating the effects of PLA2G4E overexpression in aged mice or established AD model mice would be particularly valuable. Such experiments could determine if PLA2G4E overexpression can mitigate age-related synaptic dysfunction or cognitive deficits. Positive findings in these contexts would significantly enhance the translational relevance of the research. At a minimum, a discussion of this possibility is warranted.

Thank you for this insightful comment. The role of PLA2G4E in Alzheimer's disease has indeed been extensively addressed in our previous study (Pérez-González et al.), which is cited in the current manuscript. In that work, we demonstrated that overexpression of PLA2G4E restores both memory performance and synaptic density in an AD mouse model. The present study builds upon those findings but shifts the focus toward exploring the physiological role of PLA2G4E in the central nervous system, which remains largely unknown. In response to the reviewer's suggestion, we have revised the manuscript to reduce the emphasis on Alzheimer's disease and instead highlight the contribution of PLA2G4E to synapse formation and memory under normal physiological conditions, particularly during development and aging.

4. The figures require correct annotations: please ensure accurate labeling of the "µm" unit in Figure 3D and include scale bars in Figures 3A, 3D, 3F, and 4B to comply with journal formatting standards.

Reviewer #2 (Comments to the Authors (Required)):

In this manuscript, the authors analyzed the functions of PLA2G4E in the brain using its knockout mice and showed that PLA2G4E is involved in memory-related behavior. This work provides new insights into the biological roles of PLA2G4E in brain development and cognitive function. However, the following points need to be clarified to strengthen the authors' conclusion.

Major comments

(1) LC-MS/MS analysis revealed a decrease in 2-AG in Pla2g4e knockout mice (Figure 4D). PLA2G4E is an N-acyltransferase that produces NAEs and NAEs, but not 2-AG, which is produced by different pathways. What is the authors' take on this?

We appreciate the reviewer's observation. Indeed, PLA2G4E is an N-acyltransferase involved in the production of NAEs and their derivatives, NAEs, such as AEA, and it is not directly involved in the canonical biosynthetic pathways of 2-AG. However, the observed reduction in 2-AG levels in the Pla2g4e knockout mice may reflect indirect effects on lipid metabolism or synaptic activity, rather than a direct enzymatic role. 2-AG and AEA have both distinct and overlapping roles in controlling CB1 receptor signaling, suggesting that their functions (and or levels) may be complementary or compensatory in maintaining endocannabinoid tone. For instance, altered

membrane lipid composition or impaired synapse formation resulting from the lack of PLA2G4E could influence the activity or expression of enzymes involved in 2-AG synthesis or degradation.

(2) In Figure 5D, the overexpression of PLA2G4E exhibited a significant improvement in memory. It is of interest to evaluate whether the administration of NAEs, the product of the PLA2G4E reaction, can also improve memory.

This is an important observation and it is actually addressed both in the Introduction and in the Discussion sections of the present manuscript. For instance, the following text from the introduction illustrate the reviewer's comment: "Interestingly, N-palmitoylethanolamide (PEA) and N-oleoylethanolamide (OEA) were found to mediate anti-inflammatory processes and enhance memory by activating several pathways, including those downstream of PPAR α [15, 16]. Interestingly, treatments directed to increase the levels of NAEs, such the administration of the Fatty Acid Amide Hydrolase (FAAH) inhibitor URB597 [17–19], or the direct administration of PEA or OEA improved memory in various animal models exhibiting cognitive deficits [14, 20, 21]."

Several studies using different animal models have shown that PEA administration can improve performance in memory-related tasks, possibly by enhancing synaptic plasticity or by indirectly influencing the endocannabinoid system. For example, PEA may potentiate the activity of other endocannabinoids like anandamide (AEA), which are known to play roles in learning and memory. Although clinical evidence is still limited, early findings suggest that PEA could be a promising adjunctive treatment for cognitive impairment, especially in contexts involving neuroinflammation, such as Alzheimer's disease or aging-related decline.

(3) PLA2G4E shows a bell-shaped expression profile, peaking at PND8 and declining thereafter (Figure 1A), while Figure 7A indicates peak expression at the later stage, 9 months. What is the difference?

According to other authors, who have studied mRNA expression levels or enzymatic activity, as described in Figure 1, PLA2G4E exhibits a peak of expression during the postnatal period and remaining at very low basal levels throughout adulthood (1–8 months). However, our data also show that when examining expression across the mouse lifespan, a secondary peak emerges around 8–9 months of age, followed by a subsequent decline as the animal continues to age. In fact, this biphasic expression pattern is consistent with data reported for this enzyme in the Mouse Dementia Network web server (mouseac.org). Although we currently lack a definitive explanation for this specific regulation, we hypothesize that it is related to the requirement for PLA2G4E during particularly vulnerable periods such as development and aging. Since we understand that this may lead to confusion, as was the case for the reviewer, this clarifying text has been added to the results section accompanying Figure 7.

(4) The biological functions of PLA2G4E are thought to be exerted through its reaction product, NAEs. PPAR α and CB1 are known to be the receptors for NAEs and may be involved in the results observed in this study. The authors should discuss the possible mechanism involving these and other receptors for NAEs.

We thank the reviewer for this insightful comment. It is already mentioned in the introduction that , N-palmitoylethanolamide (PEA) and N-oleoylethanolamide (OEA) were found to mediate anti-inflammatory processes and enhance memory by activating several pathways, including those downstream of PPAR α . Specifically, PEA and OEA can selectively activate PPAR α to prevent oxidative and inflammatory processes and limit cerebral injury, eventually resulting in

reduced neuronal death (Bordet et al.,2006). It is also important to consider that this phospholipase plays a crucial role in membrane trafficking, particularly within clathrin-independent endocytic and recycling pathways. These pathways are essential for the rapid, stimulus-dependent endocytosis that supports synaptic vesicle recycling. In this context, PLA2G4E may act not only through its enzymatic activity and lipid signaling products, such as NAEs, but also as a cargo protein that contributes to the organization of neuronal membrane remodeling events required for synapse formation and plasticity. Therefore, we cannot rule out either of these two mechanisms, or others, to explain its role in mediating memory formation and dendritic spine development. Minor comments

The explanations of the Figures and their legends are not clear and need improvement.

There are multiple bands in NeuN in Figure 1A and PLA2G4E in Figure 2A. The authors should specify the position of target proteins.

The NeuN antibody recognizes at least two main protein species that migrate around 45–50 kDa, appearing as a doublet in western blots of brain extracts (Lind et al., 2005, J Neurosci Res. 2005;79:295–302; Mullen et al. 1992; Development ;116:201–211).

In Figure 1C, please specify PND8, 15, and 30.

The explanation of Figure 1D is missing.

In Figure 5F, please clarify "V" and "S" above the blot.

Thank you for your comments. All of them have been addressed and incorporated into the revised version of the manuscript.

Reviewer #3 (Comments to the Authors (Required)):

This manuscript describes studies of Pla2g4e in the mouse brain, and impacts of constitutive and conditional knockouts as well as rescue of behavioral deficits by rAAV-mediated expression by peripheral to central delivery of Pla2g4e. Overall the data package is pretty diffuse, the data pieces are somewhat disparate and though they lead to some potentially interesting insight into Pla2g4e and impacts on neurons and behavior in mice, a lot of the data lacks rigor reducing the confidence of the results. The most interesting studies on rAAV overexpression rescue and inducible knockouts lack depth in terms of characterizing the in-situ effects.

General comments. No MW markers on any gel. Extra bands suggest the antibody is not specific. No evidence for key replication of data. Sex as a biologic variate is not examined. No evidence for experimental blinding in behavioural studies

We have included MW markers and the size of the bands in all the blots. Regarding the specificity of the antibody, despite the detection of multiple bands, we have demonstrated the specificity of the band at 110 KDa in Supplementary Figures S1 and S3. In Figure S1, the silencing efficiency of anti-Pla2g4e shRNA sequences was assessed in SH-SY5Y cells co-transfected with a murine PLA2G4E-expressing plasmid (pRK5-PLA2G4E) and a pSUPER recombinant vector encoding one of the three anti-Pla2g4e shRNAs. As shown in wells 3 and 4, a distinct band (110 kDa) confirms the expression of PLA2G4E following transfection with the pRK5-PLA2G4E plasmid. The intensity of this band was notably reduced in wells corresponding to cell extracts co-transfected with the plasmid and the shRNA-expressing vectors, indicating effective knockdown. In Supplementary Figure S3A, PLA2G4E expression is detectable in wild-type mice but absent in knockout (KO) mice, further supporting the antibody's specificity. Furthermore, Figure 3B confirms the

specificity of the antibody in immunohistochemistry, as the signal observed in the dentate gyrus (DG) of the hippocampus in wild-type mice is absent in knockout (KO) mice.

No behavioral differences between male and female KO mice were observed in our preliminary analyses, therefore, both sexes were used interchangeably in the experiments. Although it was missing in the methods section, all behavioral experiments were conducted with the experimenter blinded to the genotype of the animals to minimize bias. We acknowledge these comments and have clarified these points in the Methods section.

Minor note mouse Pla2g4e is being studied in most cases not human, and it should not be all caps.

Thank you for the comment. This detail has been corrected throughout the manuscript, ensuring appropriate capitalization when referring to the mouse (Pla2g4e) and human (PLA2G4E) genes.

The "spin" of the paper is on AD (at least the first sentence) but the paper really has little if anything to do with AD, it's really a study of Pla2g4e and behavior. The focus on AD might want to be shifted. Indeed, the discussion barely relates the findings to AD.

Overall, the paper is loosely written and hard to follow with respect to what was done. It would be improved by editing for conciseness and flow.

Thank you very much for your comment. Although PLA2G4E was initially identified in the context of Alzheimer's disease, your observation is appropriate and constructive. We have revised the manuscript to shift the focus toward the physiological role of this phospholipase, which is the central aim of this study. Accordingly, we have reduced the emphasis on Alzheimer's disease throughout the text, as suggested. Furthermore, the manuscript has been thoroughly revised and edited to improve its clarity and overall flow.

Specific concerns on the data and studies.

Figure 1. No data documenting the specificity of the Pla2g4e antibody is provided. No MW or full-length gels provided. If the same antibody being used as in Figure 2a this clearly has many other cross-reactive bands. This renders the IHC and IF in the figure suspect. Having two antibodies, some absorption controls, even overexpression in situ would help. Actually, the KO brains should be used as a control (from studies in Figure 4)

Despite the detection of multiple bands, the specificity of the antibody for the 110 kDa band has been demonstrated in Supplementary Figures S1 and S3. In Figure S1, the silencing efficiency of anti-Pla2g4e shRNA sequences was assessed in SH-SY5Y cells co-transfected with a murine PLA2G4E-expressing plasmid (pRK5-PLA2G4E) and a pSUPER recombinant vector encoding one of the three anti-Pla2g4e shRNAs. As shown in wells 3 and 4, a distinct band (110 kDa) confirms the expression of PLA2G4E following transfection with the pRK5-PLA2G4E plasmid. The intensity of this band was notably reduced in wells corresponding to cell extracts co-transfected with the plasmid and the shRNA-expressing vectors, indicating effective knockdown. In Supplementary Figure S3A, PLA2G4E expression is detectable in wild-type mice but absent in knockout (KO) mice, further supporting the antibody's specificity. Furthermore, Figure 3B confirms the specificity of the antibody in immunohistochemistry, as the signal observed in the dentate gyrus (DG) of the hippocampus in wild-type mice is absent in knockout (KO) mice.

We are indeed highly concerned about antibody specificity in general, and we made efforts to validate our findings using always alternative antibodies. Unfortunately, in this case and despite

the manufacturer's claims, the antibody anti-PLA2G4E from Biorbyt, catalog # orb312774) failed to detect the protein in cells transfected with a murine PLA2G4E-expressing plasmid (pRK5-PLA2G4E), therefore, we excluded this antibody from our study.

Figure 2. No control for toxicity in the shRNA experiments. As many studies the rescue experiment which is the best control is not performed (co-express a non-targeted Pla2g4e and show the effects reverse). The RNAseq data is shown but does RNA seq support reduction of Pla2g4e and Vglut1? If not, then one worries about false discovery. -log₁₀ p values are shown not a p adjusted for multiple testing. Primary cell studies must be repeated, it's too easy to have a one off from a non-healthy culture. No assessments of toxicity are provided,

Thank you for this valuable comment; the reviewer is right. One indicator of the toxicity associated with many shRNA vectors is the induction of an interferon response (Bridge et al., Nature Genetics, 2003). However, this does not appear to be the case in our study. Transcriptomic analysis revealed no evidence of such a response, as shown in the following table, where no statistically significant changes were observed in the expression of interferon-stimulated genes, including *Oas2*. Moreover, the effects observed in our experiment indicate that the suppression of Pla2g4e expression specifically affects synapse-related functions without compromising neuronal integrity. This lack of neuronal damage explains why an interferon response, as suggested by the reviewer, was not observed in our results.

ID	level	havanna	gene	chr	DB	Type	start	end	strand	logFC	AveExpr	t	P.Value	adj.P.Val	B
Ifngr2		OTTMLISG00000		chr16	HAVANA	gene	91343960	91362511	+	0.184695	5.928256	1.8873514	0.080546	0.233130931	-5.4232152
Ifnar2		OTTMLISG00000		chr16	HAVANA	gene	91169671	91202477	+	0.134409	3.284514	0.8984011	0.384535	0.603289482	-6.0861845
Ifnar1		OTTMLISG00000		chr16	HAVANA	gene	91282126	91304329	+	0.039911	6.025059	0.5109761	0.617527	0.785330283	-6.9814723
Ifngr1		OTTMLISG00000		chr10	HAVANA	gene	19467697	19485977	+	-0.03726	3.918758	-0.2435968	0.811164	0.904785649	-6.652114
Isg20I2		OTTMLISG00000		chr3	HAVANA	gene	87837621	87847903	+	0.115266	4.47009	1.1425028	0.272877	0.493767959	-6.1715674
Isg20		OTTMLISG00000		chr7	HAVANA	gene	78563172	78570144	+	0.149794	0.323472	0.421212	0.680153	0.826475383	-5.577463
Isg15		OTTMLISG00000		chr4	HAVANA	gene	156283912	156283253	-	-0.0583	-1.39584	-0.104255	0.918483	0.959466094	-5.2466693
Oas2		OTTMLISG00000		chr5	HAVANA	gene	120868398	120887918	-	0.992029	-2.96832	0.8094063	0.432106	0.644658764	-4.8088929
Oas1c		OTTMLISG00000		chr5	HAVANA	gene	120938259	120950579	-	0.296968	0.350993	0.8048631	0.434686	0.647209001	-5.3676516
Oas3		OTTMLISG00000		chr5	HAVANA	gene	120891163	120915726	-	0.504866	-2.17409	0.6951234	0.498634	0.698130461	-4.9488477
Oas12		OTTMLISG00000		chr5	HAVANA	gene	115034997	115050295	+	0.336236	-0.64393	0.5811974	0.570574	0.752437575	-5.2666353
Oas1b		OTTMLISG00000		chr5	HAVANA	gene	120950700	120962228	+	0.135923	-0.9704	0.2308798	0.820831	0.90927725	-5.3161449
Stat2		OTTMLISG00000		chr10	HAVANA	gene	128106428	128128718	+	-0.15744	4.629612	-1.4183211	0.178509	0.38190043	-5.8768361
Stat1		OTTMLISG00000		chr1	HAVANA	gene	52158999	52201024	+	0.307721	2.030744	1.0776233	0.299866	0.521643354	-5.5592974
Irf9		OTTMLISG00000		chr14	HAVANA	gene	55841028	55847487	+	-0.19495	3.301416	-1.2637658	0.227457	0.442512369	-5.7103809

ID	set	enrichment	NES	pvalue	p.adjust	qvalues	rank	Count	core_enrichment
GOBP_REGULATION_OF_RESPONSE_TO_INTERFERON_GAMMA	23	-0.1772605	-0.520410254	0.98409994	0.995551	0.857761	15893	22	Nrl1/2/3/4/5/6/7/8/9/10/11/12/13/14/15/16/17/18/19/20/21/22/23/24/25/26/27/28/29/30/31/32/33/34/35/36/37/38/39/40/41/42/43/44/45/46/47/48/49/50/51/52/53/54/55/56/57/58/59/60/61/62/63/64/65/66/67/68/69/70/71/72/73/74/75/76/77/78/79/80/81/82/83/84/85/86/87/88/89/90/91/92/93/94/95/96/97/98/99/100

The RNA-seq data support a significant reduction in Pla2g4e expression (see boxplot below), as well as in Vglut1. As described in the text and shown in Supplementary Table S2, Vglut1 (Slc17a7)

is among the most significantly downregulated genes, with a log fold change of -1.30 and an adjusted p-value of 2.02×10^{-6} .

Figure 3. This is not inhibition but knockdown. Again, no assessments of off target effects or toxicity. Again, were these experiments repeated? Can the knockdown effect be rescued with Pla2g4e co-expression. No Pla2g4e staining is shown. One might predict that the cells with decreased arbors should have lower PLA2g4e expression.

As previously explained in our response, and based on the transcriptomic analysis, treatment with shPla2g4e does not trigger an interferon response. This, combined with the specific impact on synapse-related functions—without compromising neuronal integrity—and the absence of effects when the treatment is applied at DIV10, supports the conclusion that the shRNA does not induce toxicity or broad off-target effects

The reviewer is right, it is knockdown and not inhibition. The experiment was repeated in three different cell cultures and a total of 21-24 neurons were analyzed per condition as specified in the figure legend (n =21 o 24). Indeed, the neurons analyzed for arbor complexity are those transfected with the specific shRNA plasmid or control, which also co-expresses GFP. This allows us to visualize and selectively analyze only the transfected cells, which show reduced PLA2G4E expression in the case of the shRNA- targeting PLA2G4E. Therefore, the observed reduction in dendritic arborization corresponds directly to cells with lower PLA2G4E levels. Finally, no staining was performed because the cultures were collected at DIV14, and as shown in Figure 2A, endogenous expression at this stage is already very low. Therefore, further reduction upon knockdown would not result in a detectable difference. We believe that under these conditions, performing immunostaining would not provide meaningful additional information

Figure 5. The rAAV overexpression study is perhaps the most potentially impactful data, but appears to be highly underpowered, sex as a biologic variable is not assessed, and data on in situ overexpression is not provided. The calculation of vg appears in error, or if it is correct then the vg used are almost certainly incompatible with mediating high levels of expression in the brain. Methods and text are hard to reconcile on this calculation. The lack of in situ data showing how many cells are actually expressing Pla2g4e is a major concern. The behavioural effect looks impressive but if only few cells are getting transduced and expressing Pla2g4e a confound exists. The labeling of the western blot in figure 5 is not described. What is V what is S? What about looking at synapses or other assessments of phenotypes in the brain?

Thank you for the comments. In this experiment, our goal was to perform a proof-of-concept rescue experiment in PLA2G4E knockout mice to demonstrate that the observed memory

deficits were specifically due to the absence of PLA2G4E. With respect to the consideration of sex as a biological variable, our preliminary analyses did not reveal any significant behavioral differences between male and female knockout (KO) mice. As a result, both male and female were included and used interchangeably in all the experiments to avoid sex-based bias.

We have revised the labeling of the western blot to clearly indicate the condition corresponding to each lane. We hope that this updated version improves clarity for the reader.

We thank the reviewer for raising the important point of the expression of PLA2G4E subsequent to the administration AAVP31-*PLA2G4E*. We used a dose of 2.5×10^{11} vg/kg; this value has been corrected in the manuscript. Although this can be considered a low dose, we were still able to observe transfected neurons by immunohistochemistry (as shown in the updated version of the Figure 5) and an increase in PLA2G4E expression by western blot, which was sufficient to induce a detectable phenotypic effect in this specific mouse model. While we acknowledge that only a subset of cells may be transduced and express *Pla2g4e*, it is important to note that even a limited number of functionally relevant cells can be sufficient to induce a measurable behavioral effect. This is consistent with findings in neurodegenerative diseases, where substantial behavioral phenotypes often emerge only after a significant loss of specific neuronal populations. In our case, the observed behavioral changes likely reflect the functional impact of *Pla2g4e* expression in a critical subset of neurons, rather than requiring widespread transduction. Moreover, it is well established that healthy or functionally enhanced neurons can partially compensate for the dysfunction of neighboring diseased cells, which may help explain how targeted expression in a limited population can result in a detectable effect.

In this experiment, our goal was to perform a proof-of-concept rescue experiment in PLA2G4E knockout mice to demonstrate that the observed memory deficits are specifically due to the absence of *Pla2g4e*. It is reasonable to assume that restoring *PLA2G4E* expression and having an effect in the mouse phenotype, it will have an increase in the synapses and in its products including NAEs. We believe that additional experiments, while supportive, is not critical for the main conclusion of this experiment.

Figure 7. Again, no assessment of the phenotype of the conditional and constitutive knockouts in terms of synapses, gliosis, neuronal health etc...

Besides the behavioral experiments, we have done the measurement of the synaptic markers across the first month, demonstrating a decrease in the levels of expression particularly in the frontal cortex in knockout mice (KO) compared to WT mice. While it is true that such changes were detected at certain postnatal days, they were not sustained in adult mice, at least not when assessed by western blot analysis of whole tissue lysates (Figure S6A). In addition, no significant differences between WT and KO mice were observed in dendritic spine density within the hippocampus (Figure S6B) or in electrophysiological recordings (Figure S6D-F), suggesting that these alterations in the constitutive knock out mice do not persist at the structural or functional level in the adult brain. Additionally, we performed RNA sequencing in adult animals (Figure S5), which revealed several distinct biological processes and functional differences between WT and KO mice. Given that we were unable to detect substantial differences in the constitutive KO mice using these approaches, it is likely to be even more challenging to identify such differences in the conditional model. Overall, the experiments presented in this study reveal significant differences in cognitive function between WT and KO mice, however, establishing direct molecular correlations remains challenging with the bulk tissue techniques employed.

Considering the clear behavioral differences observed, it is reasonable to assume that underlying molecular changes do exist but may be subtle or confined to specific cell types. It would therefore be of great interest to apply single-cell RNA sequencing on isolated neuronal populations, as this approach may uncover more specific or nuanced differences not detectable with the current methodologies.

Why is the FC data different for the constitutive knockout between the two studies shown (figure 7 and before)

Thank you for this observation. The apparent discrepancy in FC between the two experiments can be explained by the age of the animals used. In Figure 5, FC was conducted in adult mice at a relatively young age (4-5 months old), whereas in Figure 7, the mice were 14 months old. This age difference is critical, as cognitive decline and memory impairments become more pronounced with aging, especially in the context of a constitutive gene knockout. Therefore, the more severe memory deficits observed in the older mice in Figure 7 are consistent with the expected age-related exacerbation of the phenotype.

June 6, 2025

RE: Life Science Alliance Manuscript #LSA-2025-03323-TR

Prof. Ana Garcia-Osta
Fundación para la Investigación Médica Aplicada
Neuroscience
Pio XII, 55
Pamplona, Navarra 31008
Spain

Dear Dr. Garcia-Osta,

Thank you for submitting your revised manuscript entitled "PLA2G4E: Major Player during Brain Development and Cognitive Aging". As you will see, reviewers are overall satisfied with the changes in place and recommend publication. We concur with Reviewer 3 that the title should be adjusted. In addition the text should be revised to discuss the results on freezing as noted by this reviewer, as well as limitations on the relevance for human disease. We would be happy to publish your paper in Life Science Alliance pending these changes and final revisions necessary to meet our formatting guidelines.

- Please upload all figure files as individual ones, including the supplementary figure files; all figure legends should only appear in the main manuscript file.
- Please add ORCID ID for secondary corresponding author--they should have received instructions on how to do so.
- Please add the X and Bluesky handles of your host institute/organization as well as your own or/and one of the authors in our system.
- Please consult our manuscript preparation guidelines <https://www.life-science-alliance.org/manuscript-prep> and make sure your manuscript sections are in the correct order.
- There is a name discrepancy for one of your co-authors, Gloria Gonzalez-Aseguinolaza, in the manuscript vs. Gloria Gonzalez-Aseguinolaza in the system, please correct.
- Please add your main, supplementary figure, and table legends to the main manuscript text after the references section.
- We encourage you to revise the figure legends for Figure 1 such that the figure panels are introduced in alphabetical order.
- Please upload your Tables in editable .doc or Excel format.
- Please label the panels in the actual figure S7.
- Please add callouts for Figures 1D; S7A-B; S9A-C and Table S2 to your main manuscript text.
- Please mark the zoomed part with number 2 in Figure 4B.
- Please check that western blots are presented with gaps to separate different scans.

A. FINAL FILES:

B. MANUSCRIPT ORGANIZATION AND FORMATTING:

Sincerely,

Reviewer #1 (Comments to the Authors (Required)):

The authors have addressed all my concerns and I have no further comments.

Reviewer #2 (Comments to the Authors (Required)):

The authors adequately answered all the questions raised by this reviewer except the legend of Figure 1. Figure 1 has four data A-D, while its legend has three descriptions A-C. After this correction, I recommend this manuscript for publication.

Reviewer #3 (Comments to the Authors (Required)):

The manuscript addressed many concerns in the first review, but still remains somewhat overstated regarding conclusions and implications.

The title is overstated and should reflect what was actually done or observed.

PLA2G4E: Major Player during Brain Development and Cognitive Aging

Sounds like a review not a primary manuscript Should read that PLA2G4E is a regulator of dendritic architecture, synaptic function, and cognitive performance in mice.

Given the challenges of AAV gene therapy today due to unfortunate toxicities in trials one might want to temper the notion that an RAAV gene therapy will be a translatable way to proceed.

The relevance to human data should be discussed, as my own digging reveals that in mice Pl2g4e is expressed in the brain at an low but appreciable level but bulk rna seq and single cell reveal that in humans it is really a very low level even in AD. Further it is detected in some mouse proteomic studies but not in the massive studies done on human brain (see for example Seifar et al 2024)

Why the difference in freezing between studies in Fig 5 on the KO and fig 7? Needs some explanation. I mean no freezing is pretty unusual, Maybe there is something else going on? Are they deaf, blind, lack peripheral sensation?

Response to Reviewers

Life Science Alliance manuscript "PLA2G4E: Major Player during Brain Development and Cognitive Aging" #LSA-2025-03323-TR

We sincerely thank the editor and reviewers for the new helpful comments. In response to the concerns raised, we have prepared a detailed point-by-point response and revised the manuscript to align with the specific guidelines of the journal. We hope that this version meets the requirements for publication.

Reviewer #2:

The authors adequately answered all the questions raised by this reviewer except the legend of Figure 1. Figure 1 has four data A-D, while its legend has three descriptions A-C. After this correction, I recommend this manuscript for publication.

Thank you very much. We have revised and corrected the figure legend accordingly

Reviewer #3:

The manuscript addressed many concerns in the first review, but still remains somewhat overstated regarding conclusions and implications.

The title is overstated and should reflect what was actually done or observed. PLA2G4E: Major Player during Brain Development and Cognitive Aging Sounds like a review not a primary manuscript Should read that PLA2G4E is a regulator of dendritic architecture, synaptic function, and cognitive performance in mice.

Thank you very much for your valuable comment. We have revised the title accordingly, and we hope the new version accurately reflects the scope and significance of the work presented in this article. Revised title: Loss of PLA2G4E compromises synaptic structure and cognitive outcomes in mice

Given the challenges of AAV gene therapy today due to unfortunate toxicities in trials one might want to temper the notion that an RAAV gene therapy will be a translatable way to proceed.

Thank you for your thoughtful comment. We agree that recent challenges in AAV gene therapy highlight the need for caution, especially with systemic administration. However, it's worth noting that in diseases like Parkinson's, clinical trials using AAV vectors are already ongoing. These typically rely on intraparenchymal delivery such as the putamen, to target specific neuronal populations—such as dopaminergic neurons—. For instance, in infantile parkinsonism a similar rationale applies and the AADC enzyme is provided by an AAV with good results. These conditions often have a clear genetic cause and affect localized brain areas, making them good candidates for targeted AAV-based gene therapy.

The relevance to human data should be discussed, as my own digging reveals that in mice Pl2g4e is expressed in the brain at an low but appreciable level but bulk rna seq and single cell reveal that in humans it is really a very low level even in AD. Further it is detected in some mouse proteomic studies but not in the massive studies done on human brain (see for example Seifar et al 2024)

We thank the reviewer for the insightful observation. While it is true that public RNA-seq datasets (both bulk and single-cell) may suggest that *PLA2G4E* is expressed at low levels in the human brain, including in Alzheimer's disease samples, we would like to emphasize that these "omic" techniques may not fully capture the expression of this gene (or protein) because, we believe is expressed at very low levels, in specific subpopulations and potentially subject to transcriptional regulation.

Importantly, our group and others have previously demonstrated the expression of PLA2G4E in the brain using multiple complementary techniques, including **Western blotting, RT-PCR, and immunohistochemistry**, which confirm its presence at the protein level in both mouse and human brain samples. In particular, **Pérez-González et al., 2020 (Progress in Neurobiology)** provide clear evidence for PLA2G4E expression in human brain tissue, supporting our findings and interpretation.

Why the difference in freezing between studies in Fig 5 on the KO and fig 7? Needs some explanation. I mean no freezing is pretty unusual, Maybe there is something else going on? Are they deaf, blind, lack peripehral sensation?

Thank you for this observation. As we already explained in the response to reviewers in the previous revision, the difference is due to the age of the animals used. In Figure 5, FC was conducted in adult mice at a relatively young age (4-5 months old), whereas in Figure 7, the mice were 14 months old, with a more profound cognitive decline due to the lack of Pla2g4e plus aging. In this case, the total absence of freezing behavior may be related to the way the system records movement. It is important to note that in this context of motion tracking using a fear conditioning system, mobility is quantified over predefined time intervals (e.g., every 5 seconds). If the immobility value recorded for a given interval is zero, this does not necessarily mean that the animal was continuously moving throughout the entire period. It is possible that the animal remained still for a few seconds (e.g., 2 seconds), but did not meet the minimum duration threshold required by the system for the episode to be registered as immobility. Therefore, zero immobility values may still include brief moments of stillness that fall below the system's detection criteria.

June 23, 2025

RE: Life Science Alliance Manuscript #LSA-2025-03323-TRR

Prof. Ana Garcia-Osta
Fundación para la Investigación Médica Aplicada
Gene therapy for CNS disorders
Pio XII, 55
Pamplona, Navarra 31008
Spain

Dear Dr. Garcia-Osta,

Thank you for submitting your Research Article entitled "Loss of PLA2G4E compromises synaptic structure and cognitive outcomes in mice". It is a pleasure to let you know that your manuscript is now accepted for publication in Life Science Alliance. Congratulations on this interesting work, and many thanks for your diligence in resolving the final issues needed to adhere with LSA policy.

DISTRIBUTION OF MATERIALS:

Again, congratulations on a very nice paper. I hope you found the review process to be constructive and are pleased with how the manuscript was handled editorially. We look forward to future exciting submissions from your lab.

Sincerely,
